# A choline-releasing glycerophosphodiesterase essential for phosphatidylcholine biosynthesis and blood stage development in the malaria parasite

Abhinay Ramaprasad[1†], Paul-Christian Burda[2,3,4†], Enrica Calvani[5], Aaron J Sait[6], Susana Alejandra Palma-Duran[5], Chrislaine Withers-Martinez[1], Fiona Hackett[1], James Macrae[5], Lucy Collinson[6], Tim Wolf Gilberger[2,3,4]*, Michael J Blackman[1,7]*

[1]Malaria Biochemistry Laboratory, The Francis Crick Institute, London, United Kingdom; [2]Centre for Structural Systems Biology, Hamburg, Germany; [3]Bernhard Nocht Institute for Tropical Medicine, Hamburg, Germany; [4]University of Hamburg, Hamburg, Germany; [5]Metabolomics Science Technology Platform, The Francis Crick Institute, London, United Kingdom; [6]Electron Microscopy Science Technology Platform, The Francis Crick Institute, London, United Kingdom; [7]Faculty of Infectious and Tropical Diseases, London School of Hygiene & Tropical Medicine, London, United Kingdom

*For correspondence:
gilberger@bnitm.de (TWolfG);
Mike.Blackman@crick.ac.uk
(MJB)

†These authors contributed
equally to this work

**Competing interest:** The authors
declare that no competing
interests exist.

**Reviewing Editor:** Malcolm J
McConville, The University of
Melbourne, Australia

**Abstract** The malaria parasite *Plasmodium falciparum* synthesizes significant amounts of phospholipids to meet the demands of replication within red blood cells. De novo phosphatidyl-choline (PC) biosynthesis via the Kennedy pathway is essential, requiring choline that is primarily sourced from host serum lysophosphatidylcholine (lysoPC). LysoPC also acts as an environmental sensor to regulate parasite sexual differentiation. Despite these critical roles for host lysoPC, the enzyme(s) involved in its breakdown to free choline for PC synthesis are unknown. Here, we show that a parasite glycerophosphodiesterase (PfGDPD) is indispensable for blood stage para-site proliferation. Exogenous choline rescues growth of PfGDPD-null parasites, directly linking PfGDPD function to choline incorporation. Genetic ablation of PfGDPD reduces choline uptake from lysoPC, resulting in depletion of several PC species in the parasite, whilst purified PfGDPD releases choline from glycerophosphocholine in vitro. Our results identify PfGDPD as a choline-releasing glycerophosphodiesterase that mediates a critical step in PC biosynthesis and parasite survival.

## Editor's evaluation

This high-quality study characterizes a key enzyme in asexual red blood stages of the malaria para-sites that is used to salvage lipid precursors needed for membrane biogenesis and parasite growth in red blood cells. A previously identified glycerophosphodiesterase (PfGDPD), is shown to mediate the hydrolysis of host lyso-phosphatidycholine to generate choline, which in turn is required for para-site de novo phosphatidylcholine synthesis. Extensive analysis of the localization, growth phenotype and lipidomic profiles of PfGDPD deficient parasites indicate that this salvage pathway is essential for lipid homeostasis and asexual parasite development.

**eLife digest** Malaria kills over half a million people every year worldwide. A single-celled parasite called *Plasmodium falciparum* is responsible for the most lethal form of the disease. This malaria-causing agent is carried by mosquitos which transmit the parasite to humans through their bite. Once in the bloodstream, the parasite enters red blood cells and starts to replicate so it can go on to infect other cells.

Like our cells, *P. falciparum* is surrounded by a membrane, and further membranes surround a number of its internal compartments. To make these protective coats, the parasite has to gather a nutrient called choline to form an important building block in the membrane.

The parasite gets most of its choline by absorbing and digesting a molecule known as lysoPC found in the bloodstream of its host. However, it was unclear precisely how the parasite achieves this. To address this question, Ramaprasad, Burda et al. used genetic and metabolomic approaches to study how *P. falciparum* breaks down lysoPC.

The experiments found that mutant parasites that are unable to make an enzyme called GDPD were able to infect red blood cells, but failed to grow properly once inside the cells. The mutant parasites took up less choline and, as a result, also made fewer membrane building blocks. The team were able to rescue the mutant parasites by supplying them with large quantities of choline, which allowed them to resume growing. Taken together, the findings of Ramaprasad, Burda et al. suggest that *P. falciparum* uses GDPD to extract choline from lysoPC when it is living in red blood cells.

More and more *P. falciparum* parasites are becoming resistant to many of the drugs currently being used to treat malaria. One solution is to develop new therapies that target different molecules in the parasite. Since it performs such a vital role, GDPD may have the potential to be a future drug target.

## Introduction

The malaria parasite replicates within red blood cells (RBC). During its massive intraerythrocytic growth, the parasite produces de novo large amounts of membrane to support expansion of its parasite plasma membrane (PPM), the parasitophorous vacuole membrane (PVM) and other membranous structures of the growing parasite, as well as for formation of daughter merozoites. This extensive membrane neogenesis, which culminates in a sixfold increase in the phospholipid content of the infected RBC (*Wein et al., 2018*), requires an intense phase of phospholipid synthesis during the metabolically active trophozoite and schizont stages of the parasite life cycle.

Phosphatidylcholine (PC) is the most abundant membrane lipid in the malaria parasite, comprising 30–40% of total phospholipid (*Botté et al., 2013*; *Gulati et al., 2015*), and the parasite has evolved to produce this vital phospholipid via multiple enzymatic pathways from a variety of metabolic precursors from the host milieu (*Wein et al., 2018*; *Kilian et al., 2018*; *Figure 1*). Under normal conditions, ~89% of PC is synthesized from free choline and fatty acids via the CDP-choline-dependent Kennedy pathway that is common among eukaryotes (*Brancucci et al., 2017*; *Wein et al., 2018*). Choline is phosphorylated to phosphocholine (P-Cho) by choline kinase (CK) (*Ancelin and Vial, 1986b*), then converted to CDP-choline by a CTP:phosphocholine cytidyltransferase (CCT) (*Ancelin and Vial, 1989*) and finally condensed with diacylglycerol (DAG) by a choline/ethanolamine-phosphotransferase (CEPT) (*Vial et al., 1984*) to produce PC. PC is also generated via an alternate serine-decarboxylase-phosphoethanolamine-methyltransferase (SDPM) pathway also found in plants and nematodes, that uses host serine and ethanolamine (Eth) as precursors (*Pessi et al., 2004*). In this case, precursor P-Cho for the CDP-choline pathway is produced by triple methylation of phosphoethanolamine (P-Eth) by a phosphoethanolamine methyltransferase (PMT) (*Witola et al., 2008*). Phosphoethanolamine is in turn generated from ethanolamine sourced either from the serum or converted from serine by an unidentified parasite serine decarboxylase. Unlike yeast and mammals, the malaria parasite is unable to convert PE directly to PC through phospholipid methyl transferase activity (*Witola et al., 2008*), so the Kennedy and SDPM pathways intersect only at the point of PMT activity (*Figure 1*). PC can also be potentially produced by direct acylation of lysoPC via the Lands' cycle by an unknown lysophosphatidylcholine acyltransferase (LPCAT). However, this pathway is considered to not contribute significantly to PC synthesis under normal conditions (*Wein et al., 2018*).

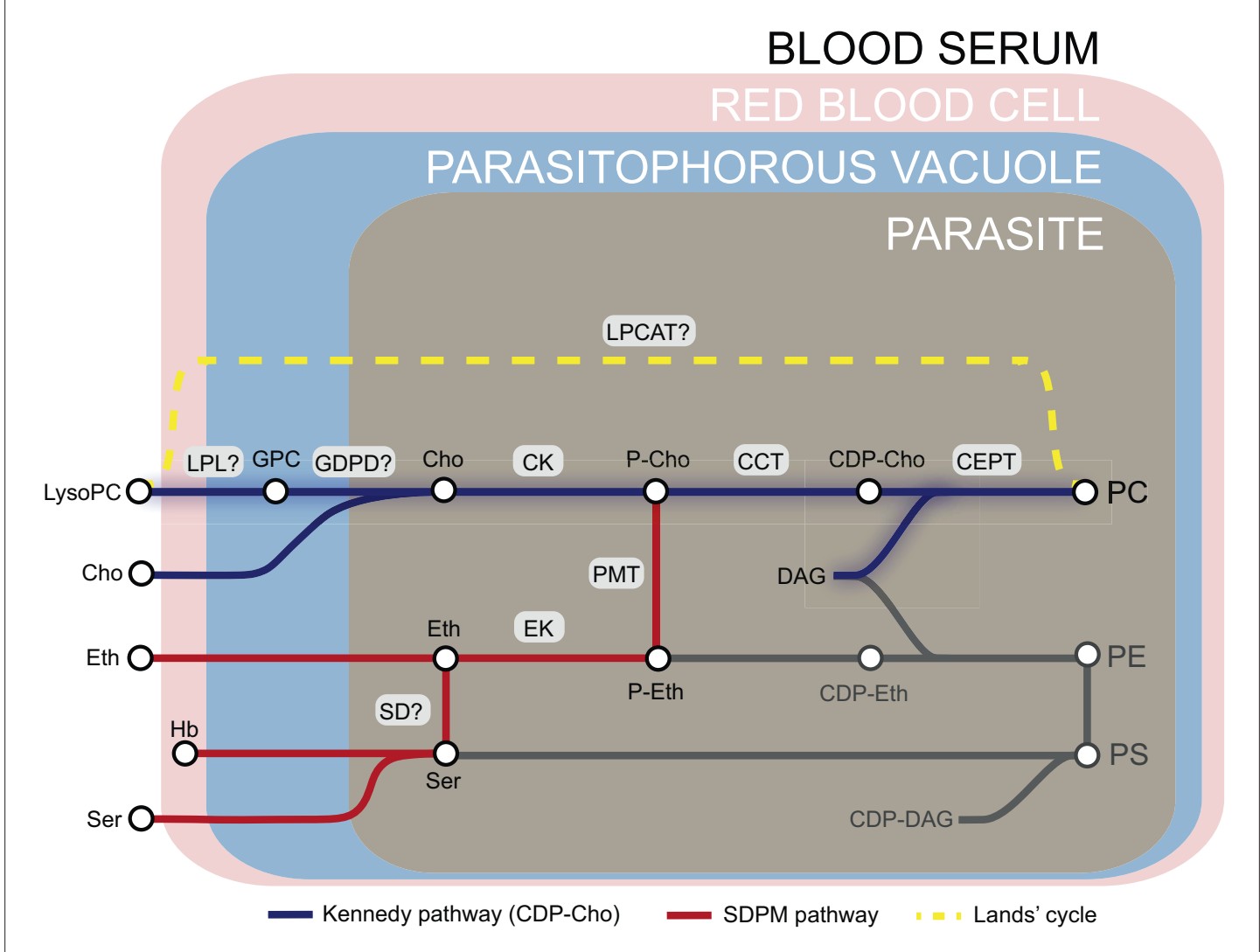

**Figure 1.** Phosphatidylcholine (PC) synthesis in malaria parasites. The CDP-choline dependent Kennedy pathway, the SDPM pathway and Lands' cycle produce PC from the metabolic precursors lysoPC, choline (Cho), ethanolamine (Eth), serine (Ser, including that obtained from digestion of haemoglobin, Hb), and fatty acids, all salvaged from the host milieu. PC is primarily produced through the Kennedy pathway using Cho sourced mainly from serum lysoPC. Breakdown of lysoPC into choline is thought to occur in the parasitophorous vacuole via a two-step hydrolysis process involving an unidentified lysophospholipase (LPL) and a glycerophosphodiesterase (GDPD; PF3D7_1406300) (this work). Other abbreviations: CCT, CTP:phosphocholine cytidyltransferase; CEPT, choline/ethanolamine-phosphotransferase; CK, choline kinase; DAG, diacylglycerol; EK, ethanolamine kinase; GPC, glycerophosphocholine; LPCAT, lysophosphatidylcholine acyltransferase; PMT, phosphoethanolamine methyltransferase; SD, serine decarboxylase. '?' indicates parasite enzymes not yet identified.

The crucial nature of PC biosynthesis for parasite survival has generated interest in this process as a potential target for antimalarial drug development (*Ancelin et al., 1985*; *Ancelin and Vial, 1986a*). Choline analogs have potent antimalarial activity (*Ancelin et al., 2003*), whilst inhibiting or disrupting enzymes in the PC synthesis pathways severely reduces or blocks intraerythrocytic growth. As examples of this, compounds that inhibit *P. falciparum* CK (PfCK) (*Serrán-Aguilera et al., 2016*) or PfCCT (*Contet et al., 2015*) in the CDP-choline pathway kill the parasite, and disruption of the PfPMT gene to block PC synthesis via the SDPM pathway results in morphological and growth defects but is not lethal (*Witola et al., 2008*). These findings suggest that the CDP-choline pathway provides the major route to PC synthesis whilst the SDPM pathway forms an important alternative route. An improved understanding of PC biosynthesis in *Plasmodium* may identify critical enzymes in the process that are potential drug targets.

The choline required for PC synthesis is primarily scavenged from host serum. Whilst free choline can cross the PPM efficiently through an unidentified carrier, choline transfer from serum into the infected RBC across the erythrocyte membrane via a parasite-induced new permeability pathway (NPP) appears rate-limiting (*Ancelin and Vial, 1989*; *Biagini et al., 2004*). Perhaps to overcome this limitation, the parasite has evolved to scavenge most of the required choline from exogenous lysoPC (*Brancucci et al., 2017*; *Wein et al., 2018*). Intriguingly, lysoPC also acts as an environmental sensor that controls sexual differentiation in *P. falciparum*. Active lysoPC metabolism into PC via the CDP-choline pathway prevents sexual commitment, while in contrast limited availability of lysoPC reduces formation of asexual progeny and triggers differentiation into the transmissible gametocyte stages (*Brancucci et al., 2017*). Metabolic labelling experiments showed that ~68% of free choline in the parasite comes from exogenous lysoPC, indicating that the majority of the lysoPC is broken down to choline before entering PC synthesis (*Brancucci et al., 2017*). However, it is unclear how and where lysoPC is converted to choline in the parasite and the enzymes involved in this process are unknown.

LysoPC breakdown to free choline requires a two-step hydrolysis reaction: deacylation of lysoPC by a putative lysophospholipase to give glycerophosphocholine (GPC) that is then hydrolysed by a glycerophosphodiester phosphodiesterase (GDPD) to generate choline and glycerol-3-phosphate (G3P) (*Figure 1*). GPC catabolism by GDPD has been shown to maintain a choline supply for CDP-choline-dependent PC biosynthesis in model eukaryotes (*Fernández-Murray and McMaster, 2005*; *Morita et al., 2016*; *Stewart et al., 2012*). Only one putative glycerophosphodiesterase gene (PfGDPD; PF3D7_1406300) has been identified in the malarial genome (*Denloye et al., 2012*). The 475-residue predicted protein product has an N-terminal secretory signal peptide and a glycerophosphodiester phosphodiesterase domain (amino acid residues 24–466; InterPro entry IPR030395), which likely adopts a triosephosphate isomerase (TIM) barrel alpha/beta fold (https://alphafold.ebi.ac.uk/entry/Q8IM31) (*Jumper et al., 2021*; *Rao et al., 2006*; *Varadi et al., 2022*). The protein shares homology with prokaryotic GDPDs and contains a characteristic small GDPD-specific insertion (residues 75–275) within the TIM barrel structure. Recombinant PfGDPD has been shown to have hydrolytic activity towards GPC and localization studies have suggested that the protein is present in the parasite digestive vacuole (in which breakdown of host haemoglobin takes place), as well as the cytoplasm and parasitophorous vacuole (PV) (*Denloye et al., 2012*). Repeated failed attempts to disrupt the PfGDPD gene (*Denloye et al., 2012*) and a genome-wide mutagenesis screen (*Zhang et al., 2018*) suggests its essentiality for asexual stage growth. However, its role in parasite phospholipid metabolism remains unknown.

Here, we used a conditional gene disruption approach combined with chemical complementation and metabolomic analysis to examine the essentiality and function of PfGDPD in asexual blood stages of *P. falciparum*. Our results show that PfGDPD catalyzes a catabolic reaction that is key for lysoPC incorporation and PC synthesis in the parasite.

## Results

### Catalytically active PfGDPD is essential for *P. falciparum* blood stage growth

To confirm the previously shown subcellular localization of PfGDPD (*Denloye et al., 2012*), we tagged the endogenous protein at its C-terminus by fusing the gene to a sequence encoding green fluorescent protein (GFP), using the selection-linked integration (SLI) system (*Birnbaum et al., 2017*; *Figure 2A*). We verified correct integration of the targeting plasmid into the PfGDPD locus by PCR (*Figure 2—figure supplement 1*). Live-cell microscopy of the resulting transgenic parasites revealed a cytoplasmic and PV localization (*Figure 2A*), supporting the results of a previous study (*Denloye et al., 2012*). Our attempts to knockout the *pfgdpd* gene using SLI-based targeted gene disruption failed, suggesting that PfGDPD fulfils an essential function for *P. falciparum* blood stage growth as previously suggested (*Denloye et al., 2012*; *Zhang et al., 2018*).

To address the essentiality and function of PfGDPD, we used a conditional gene disruption strategy to delete sequence encoding the catalytic glycerophosphodiester phosphodiesterase domain. For this, we first flanked ('floxed') the region with loxP sites using Cas9-enhanced homologous recombination in a *P. falciparum* line stably expressing DiCre, a rapamycin (RAP) inducible form of Cre recombinase (*Collins et al., 2013*; *Figure 2B* and *Figure 2—figure supplement 2*). A triple-hemagglutinin

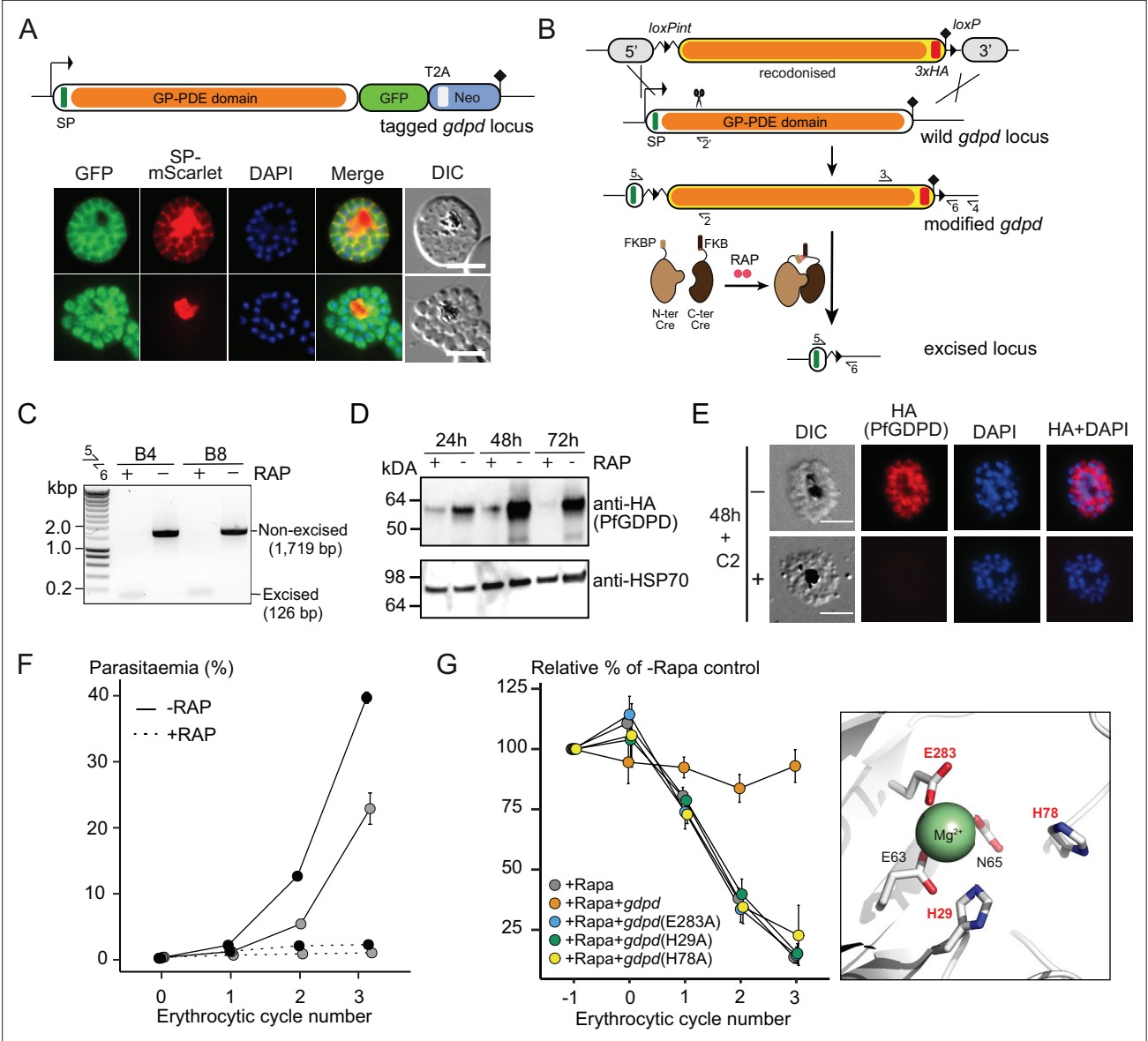

**Figure 2.** Subcellular localization and conditional ablation of PfGDPD. (**A**) Endogenous PfGDPD tagged with GFP shows dual localization in the cytosol and PV. GDPD colocalization with soluble PV marker, SP-mScarlet (**Mesén-Ramírez et al., 2019**), expressed episomally in the GDPD-GFP line is shown in mature schizonts (top) and free merozoites (bottom). Scale bars, 5 µm. (**B**) Strategy used for conditional disruption of PfGDPD in parasite line GDPD:HA:loxPint. The predicted catalytic domain (GP-PDE, glycerophosphodiester phosphodiesterase; orange) was floxed by introducing an upstream loxPint and a loxP site following the translational stop site. Sites of targeted Cas9-mediated double-stranded DNA break (scissors), left and right homology arms for homology-directed repair (5' and 3'), introduced loxP sites (arrow heads), secretory signal peptide (green), recodonized sequences (yellow), 3xHA epitope (red) and diagnostic PCR primers (half arrows 1–4) are indicated. RAP-induced DiCre-mediated excision results in removal of the catalytic domain. (**C**) Diagnostic PCR 12 hr following mock- or RAP-treatment of ring-stage GDPD:HA:loxPint parasites (representative of three independent experiments) confirms efficient gene excision. Expected amplicon sizes are indicated. (**D**) Western blots (representative of two independent experiments) showing successful RAP-induced ablation of PfGDPD expression in cycle 0 GDPD:HA:loxPint parasites sampled at 24 hr and 48 hr post invasion and cycle 1 trophozoites (72 hr). HSP70 was probed as loading control. (**E**) IFA of RAP-treated (+) and mock-treated (-) mature GDPD:HA:loxPint cycle 0 schizonts following mock- (-) or RAP-treatment (+) at ring-stage, showing that expression of PfGDPD-HA is lost following RAP treatment. Scale bar, 5 µm. (**F**) RAP-treatment results in loss of replication in two clonal lines, B4 (black) and B8 (grey), of GDPD:HA:loxPint parasites (error bars, ± SD). Data shown are averages from triplicate biological replicates using different blood sources. (**G**) Genetic complementation with an episomal, constitutively expressed mCherry-tagged PfGDPD fully restores growth of Rapa-treated GDPD:loxPint:HA:Neo-R parasites. In contrast, mutant PfGDPD alleles carrying Ala substitutions of the catalytic H29 and H78 residues or the metal-binding residue E283 do not complement. Inset, zoomed AlphaFold model of PfGDPD catalytic groove and coordinated Mg²⁺ ion, with relevant residues highlighted in red. The erythrocytic cycle when rapalog was added has been designated as cycle 0.

*Figure 2 continued on next page*

*Figure 2 continued*

The online version of this article includes the following source data and figure supplement(s) for figure 2:

**Source data 1.** Source data for plotted graphs and the full raw unedited versions of the gel and blot images shown in *Figure 2* and *Figure 2—figure supplements 1–3*.

**Figure supplement 1.** Endogenous tagging of PfGDPD.

**Figure supplement 2.** Diagnostic PCR for successful integration in GDPD:loxPint:HA line.

**Figure supplement 3.** Conditional knockout of PfGDPD expression using the SLI system.

(3xHA) epitope was simultaneously fused to the C-terminal end of the gene product, allowing confirmation of PfGDPD expression in the modified parasite line (called GDPD:loxPint:HA). Treatment of synchronous, ring-stage GDPD:loxPint:HA parasites with RAP resulted in efficient excision of floxed sequence and ablation of protein expression as determined by PCR, western blot and immunofluorescence (IFA) (*Figure 2C, D and E*). Low levels of PfGDPD-HA expression were detectable by western blot in trophozoite and schizont stages that developed throughout the erythrocytic cycle of RAP-treatment (cycle 0) (*Figure 2D*; 24 hr and 48 hr), but expression was undetectable by the beginning of the following cycle (cycle 1; 72 hr). Importantly, the RAP-treated GDPD:loxPint:HA parasites failed to proliferate, suggesting that PfGDPD is important for asexual blood stage viability of *P. falciparum* (*Figure 2F*; shown in two clonal lines, B4 and B8, of which B4 was used for all further experiments).

In parallel, we created a second conditional gene knockout line (called GDPD:loxPint:HA:Neo-R) by using the SLI system to flox a major segment of the PfGDPD catalytic domain in parasites possessing an episomally expressed DiCre recombinase (*Figure 2—figure supplement 3A* and B). As with the GDPD:loxPint:HA line, treatment with rapalog (Rapa) efficiently ablated PfGDPD expression (*Figure 2—figure supplement 3C*, D and E) and the Rapa-treated parasites displayed a severe growth defect, with proliferation being reduced by more than 85% after three erythrocytic cycles in comparison to untreated parasites (*Figure 2G*). Complementation with an episomal mCherry-tagged second copy of the gene fully restored growth of the Rapa-treated GDPD:loxPint:HA:Neo-R parasites, confirming the essentiality of PfGDPD for parasite viability (*Figure 2G*).

Like related GDPD enzymes, PfGDPD possesses two conserved predicted active site histidine residues (His29 and His78) and three metal-binding residues (Glu63, Asp65, and Glu283) that coordinate a $Mg^{2+}$ cation in the active site and are likely required for activity (*Shi et al., 2008*). Consistent with this, recombinant PfGDPD has been previously shown to display $Mg^{2+}$-dependent glycerophosphodiesterase activity (*Denloye et al., 2012*). To assess the importance of catalytic activity in PfGDPD function, we substituted both the H29 and H78 codons and a metal-binding glutamic acid (E283) codon with alanine in the complementation vector used in the GDPD:loxPint:HA:Neo-R parasites. Mutagenesis of these key residues did not alter the expression or subcellular localization of the transgenic PfGDPD:H29A:H78A:E283A protein (*Figure 2—figure supplement 3F* and G) but completely abolished rescue of parasite growth upon disruption of the chromosomal gene (*Figure 2G*). These results strongly suggest that the essential function of PfGDPD depends on its catalytic activity.

## PfGDPD is required for trophozoite development

To define in more detail the phenotypic consequences of loss of PfGDPD, intracellular development of the PfGDPD-null mutants was monitored by microscopy and flow cytometry following RAP-treatment (*Figure 3A*). Mutant parasites developed normally throughout the erythrocytic cycle of treatment (cycle 0) and were able to egress and invade fresh RBCs. However, looking more closely at the transition between cycle 0 and cycle 1, we observed that parasitaemia in the PfGDPD-null cultures was lower than controls in cycle 1 (25% vs 34%) (*Figure 3B*). Short-term replication assays under both shaking and static conditions confirmed lower fold increases in parasitaemia in the RAP-treated parasites in the transition from cycle 0 to cycle 1 (*Figure 3C*). Mean numbers of merozoites in mature cycle 0 GDPD-null schizonts were slightly lower than wild-type schizonts, perhaps contributing to the lower replication rate (*Figure 3D*). By ~24 hr into cycle 1 most of the PfGDPD-null trophozoites were developmentally arrested, at which point we also detected a decrease in DNA content (*Figure 3A*). Microscopic quantification of the various developmental stages at a range of selected time points confirmed that the majority (~88%) of PfGDPD-null mutants failed to reach schizont stage in cycle 1, instead arresting as rings or trophozoites (*Figure 3B*). The developmental defect was also

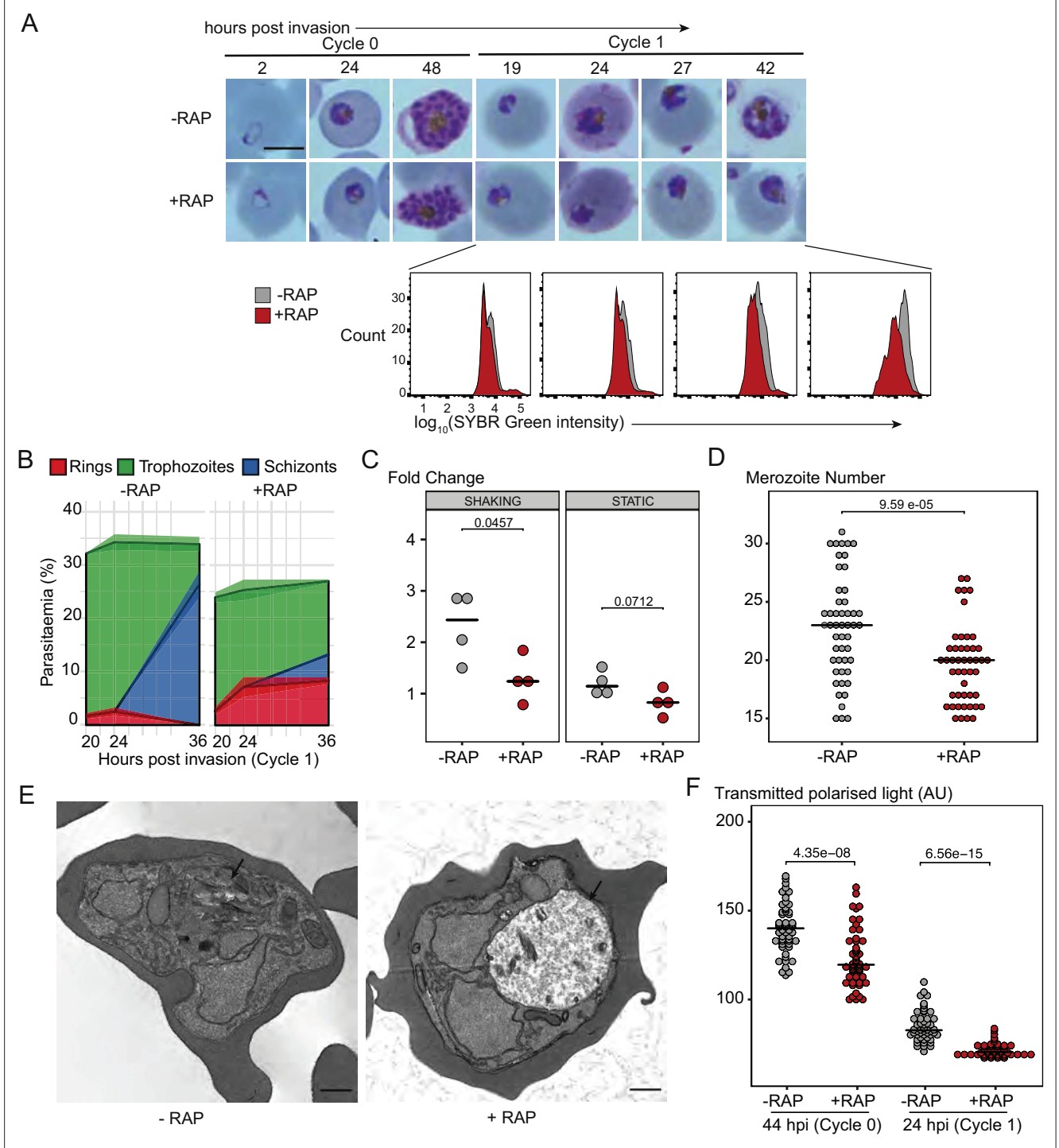

**Figure 3.** PfGDPD is essential for asexual blood stage development. (**A**) Light microscopic images of Giemsa-stained cycle 0 and 1 GDPD:HA:loxPint parasites following mock- or RAP-treatment at ring stage in cycle 0 (representative of 2 independent experiments). PfGDPD-null parasites began to exhibit defective development at around 19 hr post-invasion (19 hpi) in cycle 1, producing abnormal trophozoites. The growth defect was confirmed and quantified using flow cytometry to measure parasite DNA content. Fluorescence intensity of the SYBR Green-stained RAP-treated population (red) was detectably lower than that of the control population (grey) from 19 hr into cycle 1. Scale bar, 5 μm. (**B**) Life stage quantification of GDPD:HA:loxPint parasites at selected time points in cycle 1 (error bars, ± SD, triplicate RAP treatments) following RAP treatment of rings in cycle 0. Mock-treated parasites (DMSO) transitioned normally from trophozoite to schizont stage while RAP-treated parasites showed accumulation of abnormal ring and trophozoite forms. (**C**) PfGDPD-null parasites exhibit an invasion defect. Fold change in parasitaemia after 4 hr invasion of mock-treated (-) and RAP-treated (+) GDPD:HA:loxPint schizonts under shaking and static conditions (crossbar represents median fold change in four replicate RAP treatments

*Figure 3 continued on next page*

*Figure 3 continued*

with different blood sources; individual points represent a single replicate). (**D**) Numbers of merozoites in highly mature cycle 0 schizonts (obtained by arresting egress using the reversible egress inhibitor C2) following mock (-) or RAP-treatment (+) at ring stage. Merozoite numbers were slightly but significantly lower in PfGDPD-null parasites (crossbar represents median; n=50; Student t-test with Bonferroni adjusted p-value). (**E**) TEM micrographs of control and RAP-treated GDPD:HA:loxPint parasites allowed to mature for ~40 hr in cycle 1 in order to maximise proportions of abnormal forms. Less haemozoin formation was evident in the digestive vacuole (arrowed) of the PfGDPD-null mutants compared to mock-treated controls. Scale bar, 500 nm. (**F**) Haemozoin content of individual parasites measured as transmitted polarized light at 44 hpi in cycle 0 and 24 hpi in cycle 1. (crossbar represents median; n=50; Student t-test with Bonferroni adjusted p-value).

The online version of this article includes the following source data and figure supplement(s) for figure 3:

**Source data 1.** Source data for plotted graphs in *Figure 3*.

**Figure supplement 1.** TEM images of mock and RAP-treated GDPD:loxPint:HA parasites.

independently verified in GDPD:loxPint:HA:Neo-R parasites upon Rapa treatment (*Figure 2—figure supplement 3H and I*). Transmission electron microscopy (TEM) analysis of the growth-stalled cycle 1 trophozoites did not reveal any discernible abnormalities in morphology or membrane formation. However, we observed noticeably decreased haemozoin crystal formation in the digestive vacuole of PfGDPD-null parasites in all developmental stages (*Figure 3E* and *Figure 3—figure supplement 1*). Haemozoin content of PfGDPD-null parasites was also significantly lower than in wildtype parasites both in 44 hpi schizonts in cycle 0 (despite normal growth progression) and in 24 hpi trophozoites in cycle 1 when quantified using polarization microscopy (*Figure 3F*). Collectively, these data showed that upon RAP-treatment at ring stage to ablate PfGDPD expression, parasites were able to develop normally to schizonts in cycle 0, perhaps due to the presence of residual enzyme, but showed defective growth and reduced replication rate at the schizont stages and a definitive developmental arrest at trophozoite stages in the following cycle.

## Choline supplementation rescues the PfGDPD-null phenotype

To test for a role for PfGDPD in supplying choline to the parasite, we examined whether provision of exogenous choline could rescue the developmental defect displayed by PfGDPD-null mutants. This was indeed the case; in the presence of supraphysiological concentrations of choline (but not glycerophosphocholine, ethanolamine or serine), the RAP-treated GDPD:loxPint:HA parasites retained normal morphology (*Figure 4A*) and were able to proliferate, albeit at a ~30% slower rate than controls (*Figure 4B and C*). Confirmation that the emergent parasite population in the choline-rescued RAP-treated cultures were indeed PfGDPD-null parasites (and not residual non-excised parasites) was obtained using IFA and whole genome sequencing (*Figure 4D*). Continuous supplementation of the growth medium with choline allowed us to maintain the PfGDPD-null parasites indefinitely, and even to isolate clonal lines through limiting dilution. Growth of these clones remained completely dependent on exogenous choline (*Figure 4E*), with removal of choline resulting in the appearance of developmental defects and growth arrest within ~24 hr. Further characterization of the choline dependency using PfGDPD-null clone G1 showed that choline concentrations of 500 µM or higher were required to sustain parasite growth at near wild-type levels (*Figure 4F*). Collectively, these data clearly showed that exogenous choline can substitute for a functional PfGDPD gene, directly implicating PfGDPD in choline scavenging.

## Genetic ablation of PfGDPD results in reduced parasite levels of key structural phospholipids

To gain further insights into the essential role of PfGDPD, we compared the global phospholipid (PL) content of RAP- and mock-treated GDPD:HA:loxPint parasites. As described above, the developmental defect in synchronised RAP-treated parasites resulted in a heterogeneous population in cycle 1, ranging from developmentally arrested rings to stalled trophozoites. Because we feared that this growth arrest might itself lead to widespread metabolic dysregulation that could mask or confound changes causally associated with PfGDPD function, we initially chose to focus our PL analysis on mature cycle 0 schizonts (*Figure 5A*). At this time point, we were also able to tightly synchronise the parasite population to reduce inter-replicate variability, by allowing the schizonts to mature in the presence of the egress inhibitor C2 prior to lipid extraction. While our previous data indicated that

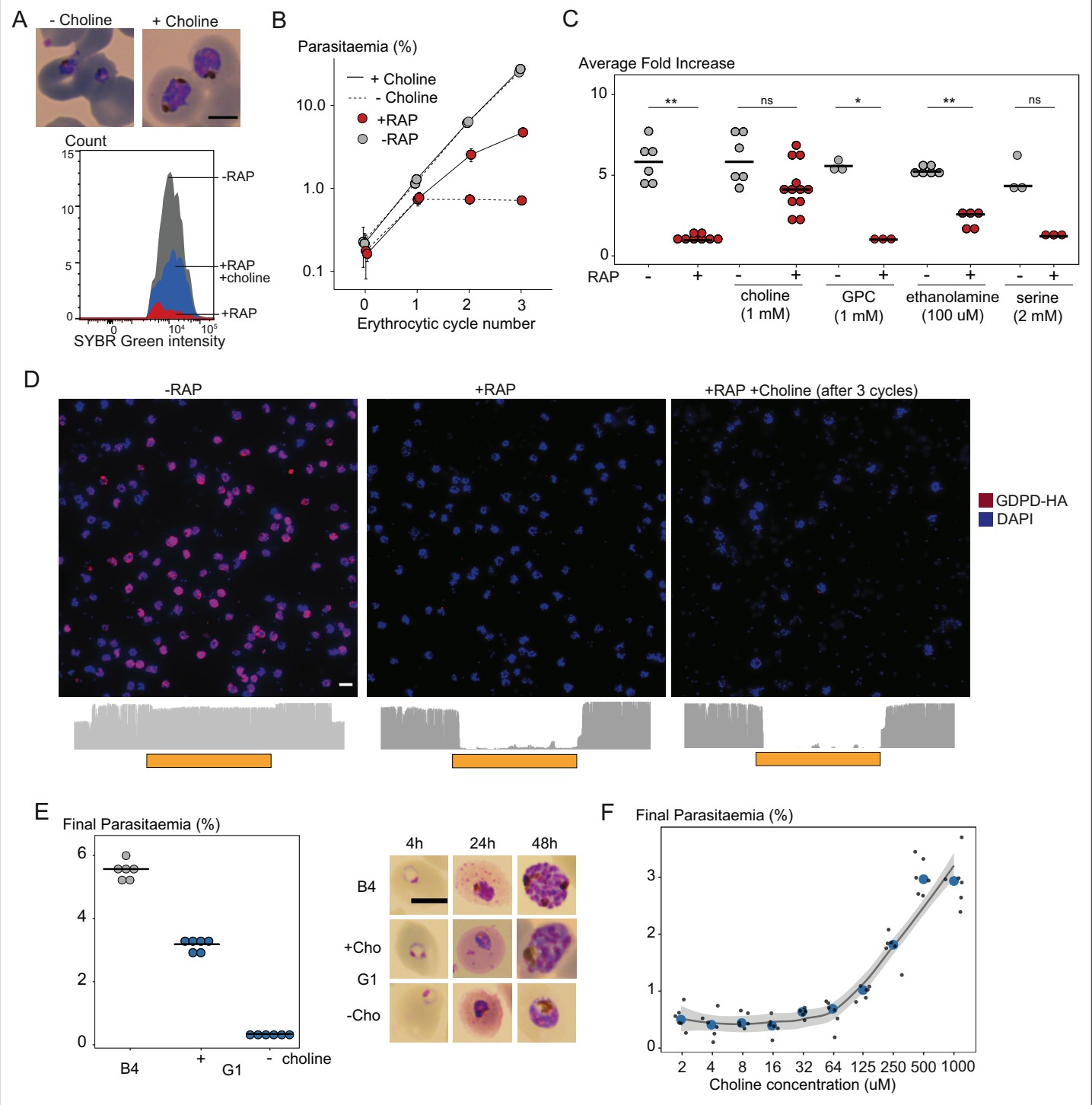

**Figure 4.** Choline supplementation rescues growth of PfGDPD-null parasites. (**A**) Morphology of PfGDPD-null trophozoites at 32 hr in cycle 1 following RAP treatment of rings in cycle 0 in the presence or absence of choline. Fluorescence intensity of SYBR Green-stained populations at the same timepoint show choline-supplemented PfGDPD-null trophozoites (blue) can surpass the developmental arrest in non-supplemented parasites. Scale bar, 5 μm. (**B**) Replication of mock- (grey) and RAP-treated (red) GDPD:HA:loxPint parasites in the presence (solid line) or absence (dashed line) of choline (error bars, ± SD, triplicate experiments with different blood sources). (**C**) Effects of supplementation with different metabolic precursors on the replication of mock- (grey) or RAP-treated (red) GDPD:HA:loxPint parasites. Mean average fold increase in parasitaemia over two erythrocytic cycles was increased by 1 mM choline to close to wild type levels (grey). In contrast, 100 μM ethanolamine effected only a marginal improvement in the replication rate while 1 mM glycerophosphocholine (GPC) and 2 mM serine had no effect. (**D**) Continuous culture of PfGDPD-null parasites enabled by choline supplementation. Top, IFA showing absence of PfGDPD-HA expression in the emergent parasite population after three erythrocytic cycles of growth in

*Figure 4 continued on next page*

*Figure 4 continued*

choline-supplemented medium (right). For comparison, parasite populations in cycle 0 following treatment are shown (left and middle). Below, genome sequencing showing RAP-induced excision of the PfGDPD gene and no evidence of the non-excised locus in the choline-supplemented emergent RAP-treated parasite population. Scale bar, 10 µm. (**E**) Confirmation of the choline dependency of the PfGDPD-null parasite clone G1. Left, parasite cultures (starting parasitaemia 0.1%) were maintained with or without 1 mM choline for two erythrocytic cycles before measuring final parasitaemia (n=6). Right, effects of choline removal on intra-erythrocytic parasite development, assessed at different time points. In all cases results are shown compared to the parental GDPD:HA:loxPint line (**B4**) without choline supplementation. Scale bar, 5 µm. (**F**) Concentration-dependence of choline supplementation on replication of the choline-dependent PfGDPD-null parasite clone G1. Parasite cultures (starting parasitaemia 0.1%) were maintained for two erythrocytic cycles in the presence of a range of choline concentrations, before final parasitaemia quantified (n=6). Black dots, individual replicates. Blue dots, mean values. Grey band, dose-dependency curve ± SD.

The online version of this article includes the following source data for figure 4:

**Source data 1.** Source data for plotted graphs in *Figure 4* and a representative microscopy image of choline-supplemented G1 line (G1+ in *Figure 4E*).

residual PfGDPD was still present at this stage, we reasoned that the reduced merozoite numbers and replication defect observed at the end of cycle 0 was indicative of a reduction in PfGDPD function that might be reflected in alterations to the PL repertoire.

Quantitative lipidome analysis detected a total of 134 PL species in both RAP- and DMSO-treated mature GDPD:HA:loxPint schizonts. Of these, we observed decreases in abundance in the RAP-treated parasites of all the major PL classes, including PC, PS, phosphatidylethanolamine (PE) and phosphatidylinositol (PI) (*Figure 5A* and *Figure 5—figure supplement 1A*). The reduction in several PC species (10 out of 22 detected species) was significant ($p < 0.05$) but less than 1.5-fold, while levels of most PE, PS, and PI species were more drastically reduced. Greater than two-fold reductions were evident in the case of seven species (PE(32:3), PE(36:5), PE(32:32), PS(34:1), PS(18:1/18:2), PS(18:1/18:1), PS(18:0/18:1), and PS(34:1)). These changes were accompanied by substantial enrichment of DAG levels in the RAP-treated parasites, with 15 out of 27 species showing significantly higher levels compared to controls.

Previous work has shown that under choline-limiting conditions the parasite can switch from the CDP-choline pathway to the SDPM pathway to produce PC (*Wein et al., 2018*). Similarly, depleted lysoPC levels cause upregulation of PfPMT (*Brancucci et al., 2018*; *Brancucci et al., 2017*; *Wein et al., 2018*). We interpreted our lipidomics results as suggesting that a similar switch occurs in PfGDPD-null parasites in order to maintain PC biosynthesis, at the expense of most of the available serine and ethanolamine pool being redirected towards PC biosynthesis, in turn resulting in lowered PE and PS production. This disturbance in precursor availability likely reduces usage of DAG, the primary backbone for glycerolipid and neutral lipid production, resulting in its accumulation. Collectively, these results indicated the onset of disruption in PL biosynthesis and were consistent with a major role for PfGDPD in this process.

## PfGDPD ablation severely reduces choline uptake from lysoPC

In view of the prior evidence that most choline scavenged by the parasite is through degradation of host serum-derived lysoPC (*Brancucci et al., 2017*), we next investigated whether PfGDPD plays a role in choline release from exogenous lysoPC. To do this, we performed isotopic lipid analysis of parasites grown in the presence of deuterium ($^2$H) choline-labelled lysoPC 16:0 ($^2$H-lysoPC) (*Brancucci et al., 2017*).

In initial experiments, RAP- and mock-treated GDPD:HA:loxPint parasites were incubated from 24 hr in cycle 0 for 14 hr in culture medium containing $^2$H-lysoPC, followed by lipid extraction and LC-MS analysis. As shown in *Figure 5B*, 20-40% of the detectable PC species were labelled with the isotope in both treatment groups. Only a small, statistically insignificant decrease was observed in the labelled proportions for 5 out of 9 PC species detected in the RAP-treated parasites compared to the controls, indicating no effects of RAP-treatment on lysoPC metabolism.

Reasoning that efficient choline uptake from lysoPC in the RAP-treated parasites might be maintained by the residual levels of PfGDPD enzyme still present in cycle 0, we next performed a similar experiment using the clonal PfGDPD-null line G1. These parasites, supported by choline supplementation, had already been maintained for multiple erythrocytic cycles (over 12 weeks) and were therefore expected to be completely devoid of PfGDPD. We maintained G1 and B4 parasite lines in choline

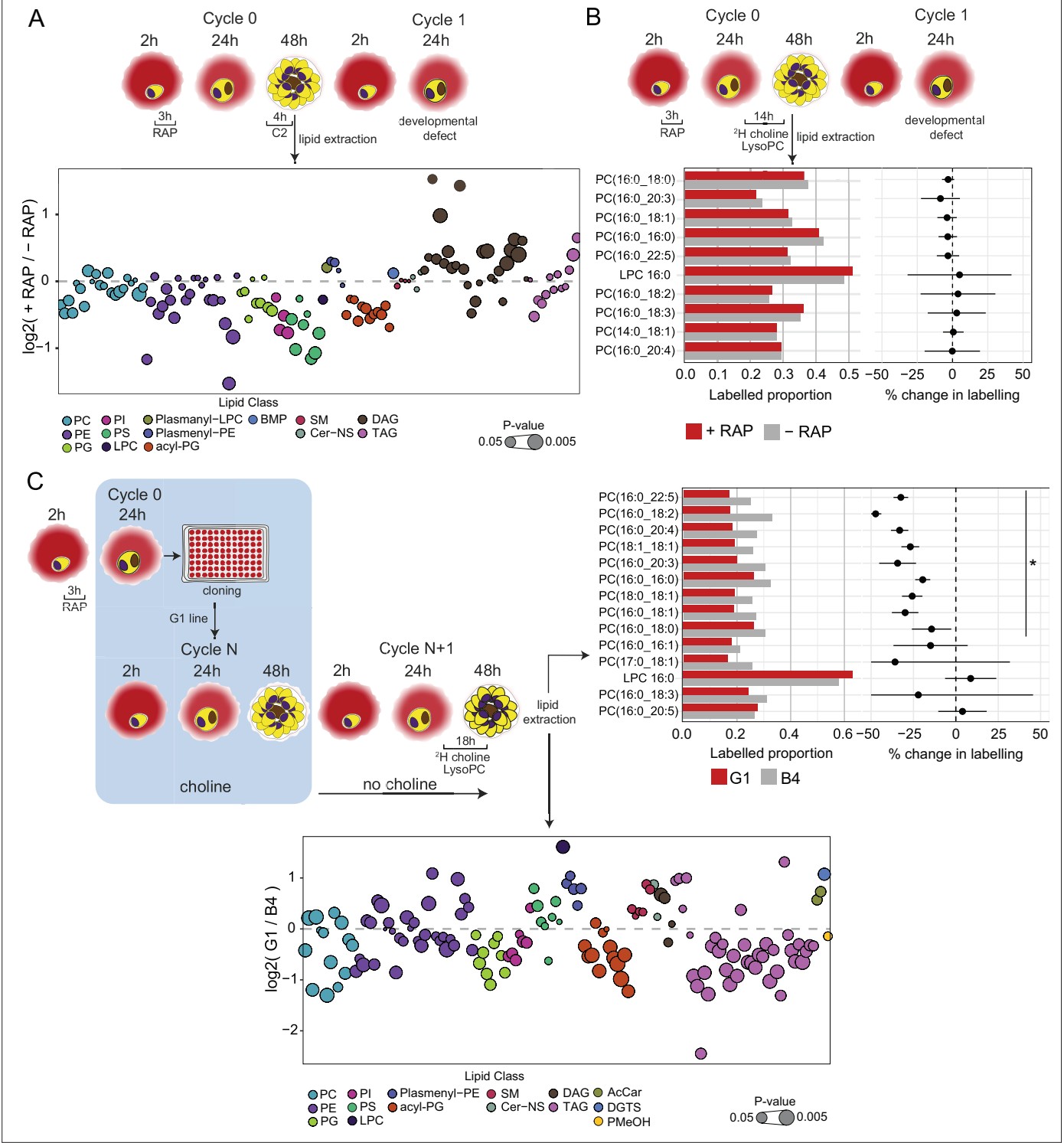

**Figure 5.** Lipidomic profiling and metabolic labelling of PfGDPD-null parasites show disruption in PC biosynthesis and choline uptake from lysoPC. (**A**) Lipidome analysis of mature cycle 0 GDPD:loxPint:HA schizonts following mock-or RAP-treatment at ring stage. The bubble plot shows the fold change in levels of various lipid species in PfGDPD-null schizonts compared to controls (3 independent biological replicates). (**B**) Metabolic labelling of mock- and RAP-treated GDPD:loxPint:HA parasites by a 14 hr incubation with $^2$H choline-labelled lysoPC 16:0 during trophozoite development. Dotplots depict percentage change in mean labelled proportions in each PC or lysoPC species (shown as bar graphs) in PfGDPD-null schizonts compared to controls across three independent biological replicates. (**C**) Metabolic labelling (top panel) and lipidome analysis (bottom panel) of PfGDPD-

*Figure 5 continued on next page*

*Figure 5 continued*

expressing GDPD:loxPint:HA (B4) and PfGDPD-null parasites (clone G1) by treatment for 18 hr with $^2$H choline-labelled lysoPC 16:0 during trophozoite development. Choline was removed from the culture medium 24 hr prior to labelling.

The online version of this article includes the following source data and figure supplement(s) for figure 5:

**Source data 1.** Source data for the plotted graphs in *Figure 5* and the PRM inclusion list (Figure5C-sourcedata3) used for identification of DGTS species in *Figure 5—figure supplement 2*.

**Figure supplement 1.** Relative peak intensities of the significantly altered lipid species.

**Figure supplement 2.** Identification of DGTS species.

**Figure supplement 3.** DNA content-based assessment of parasite development.

for one further cycle, then starved them of choline from the start of the next cycle followed by $^2$H-lysoPC treatment from 24 hr.

As shown in *Figure 5C* top panel, the choline regimen did not affect incorporation of labelled choline from $^2$H-lysoPC, as labelled proportions in the B4 controls in this experiment were comparable to mock-treated parasites used in the previous experiment without choline pre-treatment. However, a consistent and significant decrease (25–50%) in labelling of 10 out of 13 PC species was observed in the choline-starved PfGDPD-null G1 parasites compared to the B4 controls. These results strongly suggest that PfGDPD plays an important role in metabolism of exogenous lysoPC for PC synthesis.

## Loss of PfGDPD prevents de novo PC synthesis

To further explore the effects of loss of PfGDPD on phospholipid biosynthesis, we next performed a global lipidomic analysis of the metabolically labelled PfGDPD-null G1 clone parasites under choline-starved conditions, comparing them to PfGDPD-expressing B4 parasites (*Figure 5C* bottom panel and *Figure 5—figure supplement 1B*). This revealed large-scale changes in phospholipid and neutral lipid species in the choline-starved G1 parasites. Several PC species (7 out of 14 PC species) were significantly depleted in choline-starved G1 parasites, with three species - PC(16:0/18:3), PC(16:0/16:1) and PC(16:0/20:5) - decreased in abundance more than two-fold. This was accompanied by a concomitant increase in levels of lysoPC (LPC) species, LPC(16:0) and LPC(18:0). Similar to what we observed in PfGDPD-null schizonts, we noticed significant accumulation of DAG species (DAG(18:0/20:4) and DAG(18:0/22:5)), pointing to Kennedy pathway dysfunction.

Several phosphatidylglycerol (PG(18:1/18:2), PG(18:1/18:1) and PG(36:3)) and acyl-phosphatidylglycerol (acyl-PG) species were also notably depleted. All PI species detected were significantly depleted in choline-starved G1 parasites, as observed in PfGDPD-null schizonts. On the other hand, in contrast to the PfGDPD-null schizonts, most PE and PS species were unchanged between the mutant G1 parasites and the B4 controls. Another notable feature was the significant depletion of several species of TAG in choline-starved G1 parasites, with 8 species showing almost two to fivefold decrease in abundance. We also observed a twofold increase in the abundance of the betaine lipid diacylglyceryl-N,N,N-trimethylhomoserine (DGTS) in G1 parasites (*Figure 5—figure supplement 1B*). This unusual lipid, which functions as a substitute for PC in certain algal species including *Chlamydomonas reinhardtii* (*Giroud et al., 1988*; *Sato et al., 1995*), has not been previously detected in malaria parasites. We were able to match the MS/MS spectra of three DGTS species (DGTS(34:1), DGTS(35:1) and DGTS(38:2)) in our samples to that of a commercially available DGTS standard (DGTS(32:0)) thus confirming correct identification of the species (*Figure 5—figure supplement 2*). Taken together, these changes indicate major disruption in PC and lipid biosynthesis following ablation of PfGDPD.

## Purified PfGDPD releases choline from GPC

Our genetic and metabolomic data suggested that PfGDPD plays a role in the generation of free choline from lysoPC. As indicated in *Figure 1*, this pathway likely involves at least two catabolic steps: deacylation of lysoPC by a putative lysophospholipase to produce GPC, then hydrolysis of the GPC to generate choline and G3P. Recombinantly expressed PfGDPD has previously been shown to have magnesium-dependent hydrolytic activity against GPC to produce G3P (*Denloye et al., 2012*). Reasoning that PfGDPD likely catalyzes the second step in this pathway, we exploited the 3xHA epitope tag introduced into the PfGDPD protein in the GDPD:HA:loxPint parasite line to directly examine whether affinity-purified parasite-derived PfGDPD-HA has the capacity to release choline from GPC.

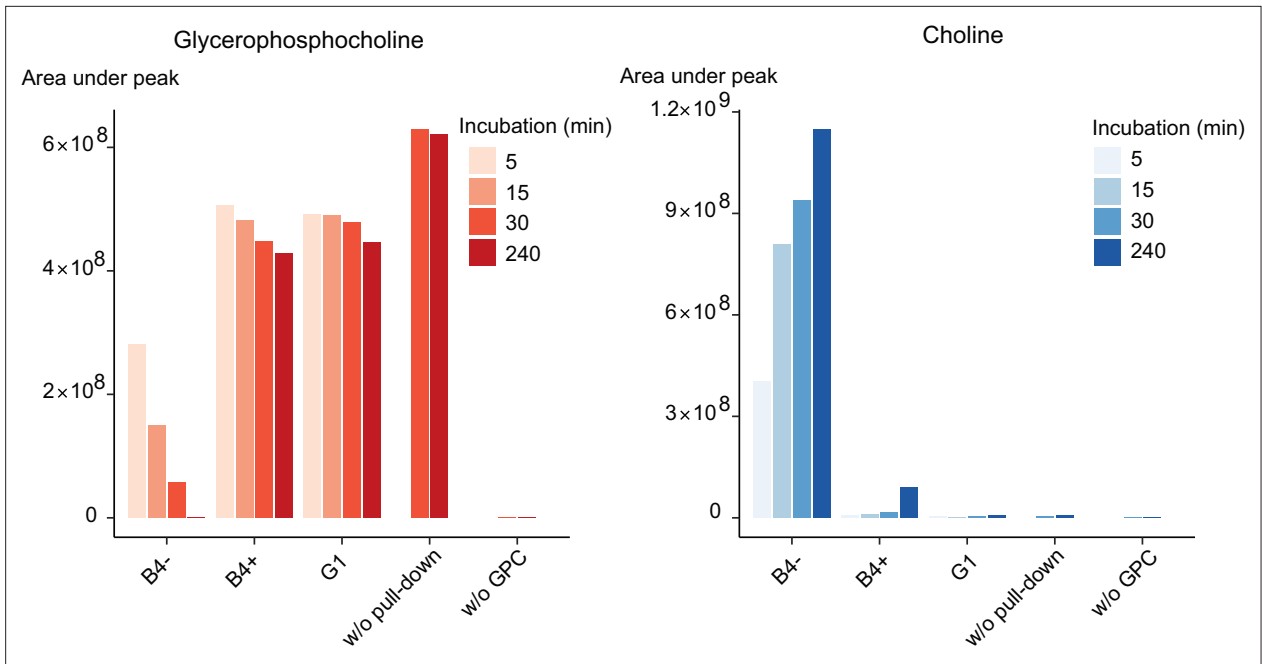

**Figure 6.** Purified PfGDPD releases choline from GPC. GPC and choline content in enzymatic reactions set up with affinity-purified GDPD-HA from similar numbers of mock- (B4-) and RAP-treated (B4+) GDPD:loxPint:HA parasites or the GDPD-null clonal parasite line (**G1**). Pulled-down samples were incubated with 10 mM GPC in a reaction buffer containing 10 mM $MgCl_2$ for different durations at 37°C. Reactions without pulled-down fraction or GPC substrate were included as controls.

The online version of this article includes the following source data and figure supplement(s) for figure 6:

**Source data 1.** Source data for plotted graph in *Figure 6* and full raw unedited versions of the blot images shown in *Figure 6—figure supplement 1*.

**Figure supplement 1.** Affinity purification of PfGDPD-HA.

**Figure supplement 2.** In silico substrate docking in PfGDPD model.

Good yields of PfGDPD-HA were obtained from saponin lysates of mock-treated GDPD:HA:loxPint schizonts while residual or undetectable levels were obtained from RAP-treated GDPD:HA:loxPint parasites and the GDPD-null clonal line, respectively (*Figure 6—figure supplement 1*). Incubating immobilised PfGDPD-HA pulled-down from control GDPD:HA:loxPint parasites with GPC resulted in the time-dependent appearance of choline with a concomitant decrease in GPC levels (*Figure 6*). As expected, the rate of choline appearance was greatly decreased using pull-downs from PfGDPD-null parasites (RAP-treated GDPD:HA:loxPint parasites or the G1 clone). These results provide direct evidence that PfGDPD can release choline from GPC.

## Loss of PfGDPD does not induce sexual differentiation

PC levels regulate sexual commitment in malaria parasites, and a block in PC synthesis through the Kennedy pathway in lysoPC-depleted conditions can induce sexual differentiation (*Brancucci et al., 2017*). Because the parental parasite line used for most of our work (3D7) is intrinsically defective in gametocytogenesis, we examined the consequences of PfGDPD disruption in a gametocyte-producing NF54 parasite line (GDPD:HA:loxPint_NF54, *Figure 7—figure supplement 1*). This showed that PfGDPD is essential for parasite growth in NF54 parasites (*Figure 7A*), with gene disruption producing a similar developmental defect at trophozoite stage as that observed in the GDPD:HA:lox-Pint line (*Figure 7B*). There was no detectable induction of gametocyte formation. This result implies that the loss of PfGDPD causes a severe block in PC synthesis resulting in the death of asexual parasites before they get to form gametocytes.

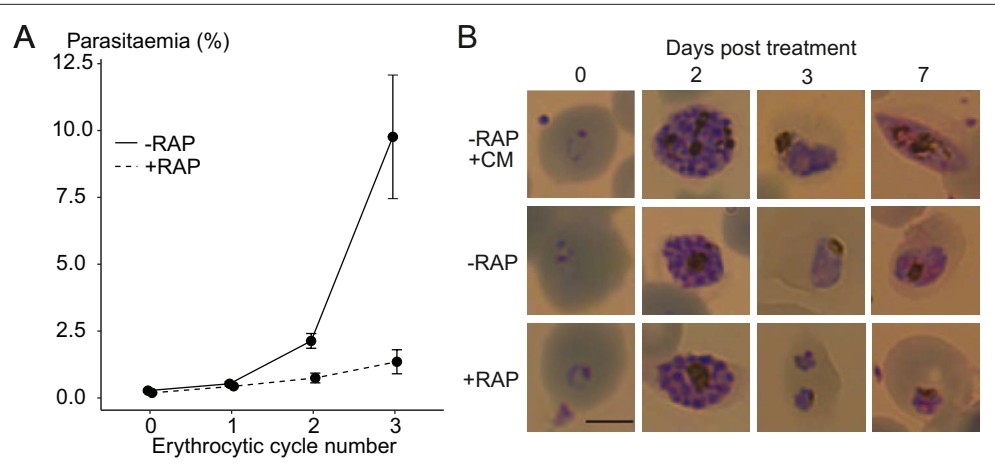

**Figure 7.** Ablation of GDPD expression does not induce sexual differentiation. (**A**) Replication of mock- (solid line) and RAP-treated (dashed line) clonal line of GDPD:loxPint:HA$_{NF54}$ parasites over three erythrocytic cycles (error bars, ± SD). Data shown are averages from triplicate biological replicates using different blood sources. (**B**) Light microscopic images of Giemsa-stained GDPD:loxPint:HA$_{NF54}$ parasites at days 0, 2, 3 and 7 post treatment with conditioned media (-RAP+CM, known to induce sexual commitment), DMSO (-RAP) or rapamycin (+RAP). Gametocyte stages were apparent from day 6–7 in cultures treated with conditioned media while DMSO-treated cultures showed normal asexual stage progression and RAP-treated cultures showed development-stalled trophozoite stages from day 3. Images are representative of three independent treatments. Scale bar, 5 μm.

The online version of this article includes the following source data and figure supplement(s) for figure 7:

**Source data 1.** Source data for plotted graph in *Figure 7A* and full raw unedited versions of the gel image shown in *Figure 7—figure supplement 1*.

**Figure supplement 1.** Diagnostic PCR for successful integration in GDPD:loxPint:HA$_{NF54}$ line.

## Discussion

Mature human RBCs are highly streamlined, terminally differentiated cells that lack a nucleus or other internal membranous organelles, with no protein synthesis machinery and limited metabolic capacity. Since erythrocytic growth of the malaria parasite requires extensive membrane biogenesis, de novo PC synthesis by the parasite is essential. Host serum lysoPC, well established as the main precursor for PC synthesis in the parasite (*Brancucci et al., 2017*; *Wein et al., 2018*), can enter the parasitized erythrocyte efficiently through rapid exchange (*Dushianthan et al., 2019*) after which it may be transported into the parasite by exported phospholipid transfer proteins (*van Ooij et al., 2013*). However, the enzymes involved in the generation of choline from lysoPC have been unknown. Here we have established PfGDPD as an essential player in this process.

Ablation of PfGDPD expression produced a phenotypic defect during trophozoite development as either a direct or general stress response to disrupted PC synthesis, similar to that observed upon inhibition of other enzymes involved in CDP-choline pathway (*González-Bulnes et al., 2011*; *Serrán-Aguilera et al., 2016*; *Contet et al., 2015*). Unsurprisingly for an enzyme playing a key role in membrane biogenesis, previous transcriptional studies have shown that PfGDPD is expressed early in the erythrocytic cycle. Our observation that DiCre-mediated disruption of PfGDPD results in parasites that undergo normal development in the erythrocytic cycle of RAP-treatment (cycle 0) is therefore likely due to the presence of residual enzyme produced early during ring stage development (prior to gene excision) and persisting throughout that cycle. In support of this, PfGDPD-null parasites that were rendered completely devoid of the enzyme through propagation for several cycles in choline-supplemented media exhibited maturation defects within ~24 hr of choline removal.

We infer that PfGDPD acts as a glycerophosphocholine phosphodiesterase and releases choline from the intermediary GPC during lysoPC breakdown from the following findings. First, site-directed mutagenesis showed that PfGDPD retains the same essential catalytic sites and metal-ion dependency as a bacterial choline-releasing GDPD enzyme (*Shi et al., 2008*). Second, supraphysiological concentrations of choline rescued the PfGDPD-null development and growth defect. Third, PC levels

and incorporation of choline from lysoPC into PC are reduced in PfGDPD-null parasites. And fourth, PfGDPD affinity-purified from parasite lysate released choline from GPC in vitro.

Previous work has shown that supplying an excess of choline can rescue the growth of parasites in lysoPC-deprived conditions (*Witola et al., 2008*; *Wein et al., 2018*). High concentrations of exogenous choline partially surpass the bottleneck in choline transport across the red blood cell membrane (*Biagini et al., 2004*) and increase choline influx through infected RBCs (*Kirk et al., 1991*). LysoPC-deprived parasites can readily take up this choline and use it for PC synthesis as shown by previous metabolic labelling studies (*Brancucci et al., 2017*). The impact of exogenous choline was well pronounced in our PfGDPD-null mutants as they completely failed to survive in the absence of choline but achieved near-wild type growth rates in high choline concentrations. This complete dependence on exogenous choline despite the abundant presence of lysoPC (amounting to ~17% of total lipid content *Garcia-Gonzalo and Izpisúa Belmonte, 2008*) in the Albumax II-supplemented culture media used in our studies or supplementation with GPC, is a further confirmation that ablating PfGDPD function interferes with choline acquisition from lysoPC and mimics a lysoPC-deprived state.

As well as its effects on PC levels, ablation of PfGDPD reduced both PE and PS content in the parasite, likely due to redirection of ethanolamine and serine precursors into PC synthesis. Consistent with this, ethanolamine supplementation marginally improved growth of PfGDPD-null parasites. Loss of PfGDPD also reduced – but did not eliminate – incorporation of lysoPC-derived choline. This suggests that alternate pathways such as direct lysoPC acylation into PC via the Lands' cycle, two acylation steps to convert GPC to PC or the SDPM pathway may contribute to PC synthesis under choline-starved conditions. However, our results strongly suggest that PfGDPD-mediated choline release from lysoPC remains the primary, indispensable pathway to meet the choline requirements of the intraerythrocytic parasite.

The observed depletion of TAG in choline-starved PfGDPD-null parasites was unexpected since there is no previously reported role for a glycerophosphodiesterase in neutral lipid metabolism. An intense phase of de novo TAG biosynthesis from DAG is known to accompany trophozoite development (*Palacpac et al., 2004*; *Vielemeyer et al., 2004*), followed by a rapid hydrolysis of TAG in mature schizonts resulting in localized release of fatty acids essential for merozoite maturation prior to egress (*Gulati et al., 2015*). Our results could simply reflect the different developmental stages of the choline starved G1 and B4 parasites at the point of lipid extraction (*Figure 5—figure supplement 3*) as a result of the developmental arrest that inevitably occurs in G1 parasites upon choline deprivation. However, the changes in lipid levels (PC in particular) that we observe here are more drastic than the normal temporal dynamics of these lipids during blood stage progression (*Gulati et al., 2015*). This suggests that the perturbations are to an extent indeed the result of loss of PfGDPD. A block in PC synthesis can either cause the depletion of the PC-derived DAG pool and block TAG synthesis, or lead to increased metabolism of TAG to feed fatty acids into the compensatory Lands' cycle that acylates lysoPC to PC (*Caviglia et al., 2004*; *Moessinger et al., 2014*). Studies in other organisms increasingly show a bidirectional link between TAG and PC synthesis in which PC-derived DAG is used for TAG synthesis and TAG-derived fatty acids are used for synthesis of PC through the Lands' cycle (*Bates and Browse, 2011*; *Caviglia et al., 2004*; *Moessinger et al., 2014*; *Soudant et al., 2000*; *van der Veen et al., 2012*). The reduced haemozoin formation in GDPD-null parasites could also be a result of this decrease in TAG levels. Neutral lipids have been suggested to play a role in the parasite haem detoxification pathway and promote haemozoin formation (*Hoang et al., 2010*). Indeed it was recently shown that knockdown of a *P. falciparum* lysophospholipase results in reduced TAG levels, reduced haemozoin formation and a block in trophozoite development (*Asad et al., 2021*).

Our lipidomics analysis identified the betaine lipid DGTS, prevalently found in photosynthetic organisms like green algae, mosses and ferns (*Giroud et al., 1988*; *Sato et al., 1995*). DGTS has also been detected in the lipidome of *Chromera velia* and *Vitrella brassicaformis*, the algal ancestors of apicomplexan parasites (*Tomčala et al., 2017*) but has not been previously reported in *Plasmodium*. Accumulation of DGTS in choline-starved PfGDPD-null parasites is strikingly reminiscent of reports of DGTS synthesis as a non-phosphorous substitute for PC in fungi and marine bacteria under phosphate- or choline-starved conditions (*Geiger et al., 2013*; *Riekhof et al., 2014*; *Sebastián et al., 2016*; *Senik et al., 2015*). The methyl donor S-adenosyl methionine (SAM) is capable of providing both the homo-serine moiety and the methyl groups to produce DGTS from DAG (*Moore et al., 2001*). Enzymes involved in SAM synthesis are upregulated in lysoPC-limiting conditions and diversion of

SAM pools from histone methylation towards compensatory PC biosynthetic pathways is the primary link between PC metabolism and sexual differentiation in *P. falciparum* (**Harris et al., 2022**). It is therefore tempting to speculate that blocking PC biosynthesis in *P. falciparum* triggers a compensatory pathway that produces DGTS as a functional substitute for PC in the parasite.

Based on protein localisation, ligand docking and sequence homology analyses, we can further speculate regarding aspects of PfGDPD function that we have not explored in this study. It has been previously suggested that the gene could use alternative start codons via ribosomal skipping to produce distinct PV-located and cytosolic variants of the protein (**Denloye et al., 2012**). PfGDPD could potentially perform similar functions in both compartments by facilitating the breakdown of exogenous lysoPC both within the PV and within the parasite cytosol (**Brancucci et al., 2017**). This scale of enzyme activity may be essential for the parasite to meet its choline needs, given the high levels of PC synthesis during parasite development and its crucial importance for intraerythrocytic membrane biogenesis. PfGDPD may also have other roles during asexual stages such as temporal and localised recycling of intracellular PC or GPC by the PfGDPD fraction expressed in the cytosol. Finally, our ligand docking simulations do not rule out additional catalytic activity towards other glycerophosphodiester substrates such as glycerophosphoethanolamine and glycerophosphoserine (***Figure 6—figure supplement 2A, B***). Further investigation that involves variant-specific conditional knockout of the *gdpd* gene could help us further dissect the role of PfGDPD in the parasite. Orthologs of PfGDPD form a *Haematozoan*-specific ortholog group (OG6_139464 in OrthoMCL DB release 6.4) that encompasses only blood parasites that have an intra-erythrocytic stage, that is the genera *Babesia*, *Theileria*, *Plasmodium* and *Hepatocystis* (***Figure 6—figure supplement 2C***). We speculate that this entire ortholog group could have a conserved role in choline acquisition for the critical process of PC biosynthesis to support an intra-erythrocytic lifestyle. PC biosynthesis is the main biosynthetic process in blood stages of *Babesia* (**Florin-Christensen et al., 2000**) and compounds that inhibit PC biosynthesis have been shown to have anti-parasite activity against *Babesia* and *Theileria* blood stages (**AbouLaila et al., 2014**; **Gopalakrishnan et al., 2016**; **Maji et al., 2019**; **Richier et al., 2006**).

In conclusion, we have demonstrated that a malaria parasite choline-releasing glycerophosphodiesterase catalyses a critical step in choline acquisition from exogenous lysoPC. Since PfGDPD-mediated procurement of choline is indispensable for normal PC biosynthesis and asexual blood stage development in the parasite, it may represent a potential new drug target.

## Methods

**Key resources table**

| Reagent type (species) or resource | Designation | Source or reference | Identifiers | Additional information |
|---|---|---|---|---|
| Gene (*Plasmodium falciparum*) | PfGDPD | PlasmoDB (https://plasmodb.org) | PF3D7_1406300 | *P. falciparum* GDPD gene |
| Genetic reagent (*P. falciparum*) | GDPD:HA:loxPint | This paper | | For inducible disruption of PfGDPD in B11 line. Line maintained in and available from Blackman lab, Francis Crick Institute. |
| Genetic reagent (*P. falciparum*) | GDPD:HA:loxPint$_{NF54}$ | This paper | | For inducible disruption of PfGDPD in NF54::DiCre line. Line maintained in and available from Blackman lab. |
| Genetic reagent (*P. falciparum*) | G1 | This paper | | Clonal GDPD-null line supplemented with choline. Line maintained in and available from Blackman lab. |
| Genetic reagent (*P. falciparum*) | GDPD:GFP | This paper | | Endogenous tagging of PfGDPD with GFP. Line maintained in and available from Gilberger lab at Centre for Structural Systems Biology, Hamburg. |

*Continued on next page*

*Continued*

| Reagent type (species) or resource | Designation | Source or reference | Identifiers | Additional information |
|---|---|---|---|---|
| Genetic reagent (*P. falciparum*) | GDPD:GFP +<sup>Epi</sup>SP-mScarlet | This paper | | Endogenous tagging of PfGDPD with GFP. Episomal expression of PV marker protein SP-mScarlet. Line maintained in and available from Gilberger lab. |
| Genetic reagent (*P. falciparum*) | GDPD:loxPint:HA:Neo-R | This paper | | For inducible disruption of PfGDPD. Generated using SLI system. Line maintained in and available from Gilberger lab. |
| Cell line (*P. falciparum*) | GDPD:loxPint:HA:Neo-R+<sup>Epi</sup>NMD3:GDPD-mCherry | This paper | | For inducible disruption of PfGDPD. Episomal expression of GDPD-mCherry. Line maintained in and available from Gilberger lab |
| Cell line (*P. falciparum*) | GDPD:loxPint:HA:Neo-R+<sup>Epi</sup>NMD3:GDPD(H29A)-mCherry | This paper | | For inducible disruption of PfGDPD. Episomal expression of GDPD(H29A)-mCherry. Line maintained in and available from Gilberger lab |
| Cell line (*P. falciparum*) | GDPD:loxPint:HA:Neo-R+<sup>Epi</sup>NMD3:GDPD(H78A)-mCherry | This paper | | For inducible disruption of PfGDPD. Episomal expression of GDPD(H78A)-mCherry. Line maintained in and available from Gilberger lab |
| Cell line (*P. falciparum*) | GDPD:loxPint:HA:Neo-R+<sup>Epi</sup>NMD3:GDPD(E283A)-mCherry | This paper | | For inducible disruption of PfGDPD. Episomal expression of GDPD(E283A)-mCherry. Line maintained in and available from Gilberger lab |
| Cell line (*P. falciparum*) | B11 | *Perrin et al., 2018* | | DiCre-expressing 3D7 parasite line. Maintained in and available from Blackman lab, Francis Crick Institute. |
| Cell line (*P. falciparum*) | NF54::DiCre | *Tibúrcio et al., 2019* | | DiCre-expressing NF54 parasite line. Maintained in and available from Treeck lab, Francis Crick Institute. |
| Transfected construct (*P. falciparum*) | pCas9_1406300_gRNA01 | This paper | | Cas9-targeting plasmid for producing GDPD:loxPint:HA line. Available from Blackman lab. |
| Transfected construct (*P. falciparum*) | pCas9_1406300_gRNA02 | This paper | | Cas9-targeting plasmid for producing GDPD:loxPint:HA line. Available from Blackman lab. |
| Transfected construct (*P. falciparum*) | pREP-GDPD | This paper | | Repair plasmid for producing GDPD:loxPint:HA line. Available from Blackman lab. |
| Transfected construct (*P. falciparum*) | pSLI-PF3D7_1406300-GFP-GlmS-WT | This paper | | GFP-tagging construct for producing GDPD:GFP line. Available from Gilberger lab. |
| Transfected construct (*P. falciparum*) | pSLI-PF3D7_1406300-TGD | This paper | | SLI-based construct for testing essentiality of PfGDPD. Available from Gilberger lab. |
| Transfected construct (*P. falciparum*) | pSLI- PF3D7_1406300-loxPint:HA:T2A:Neo | This paper | | SLI-based construct for producing GDPD:loxPint:HA:Neo-R line. Available from Gilberger lab. |
| Transfected construct (*P. falciparum*) | pSkipFlox | *Birnbaum et al., 2017* | | Plasmid for ectopic expression of DiCre in GDPD:loxPint:HA:Neo-R line. |

*Continued*

| Reagent type (species) or resource | Designation | Source or reference | Identifiers | Additional information |
|---|---|---|---|---|
| Transfected construct (*P. falciparum*) | pNMD3:PF3D7_1406300-mCherry-DHODH | This paper | | Gene complementation vector for GDPD:loxPint:HA:Neo-R line leading to episomal expression of GDPD-mCherry. Available from Gilberger lab. |
| Transfected construct (*P. falciparum*) | pNMD3:PF3D7_1406300(H29A)-mCherry-DHODH | This paper | | Gene complementation vector for GDPD:loxPint:HA:Neo-R line leading to episomal expression of GDPD(H29A)-mCherry. Available from Gilberger lab. |
| Transfected construct (*P. falciparum*) | pNMD3:PF3D7_1406300(H78A)-mCherry-DHODH | This paper | | Gene complementation vector for GDPD:loxPint:HA:Neo-R line leading to episomal expression of GDPD(H78A)-mCherry. Available from Gilberger lab. |
| Transfected construct (*P. falciparum*) | pNMD3:PF3D7_1406300(E283A)-mCherry-DHODH | This paper | | Gene complementation vector for GDPD:loxPint:HA:Neo-R line leading to episomal expression of GDPD(E283A)-mCherry. Available from Gilberger lab. |
| Transfected construct (*P. falciparum*) | SP-mScarlet | **Mesén-Ramírez et al., 2019** | | PV marker for GDPD:GFP line. |
| biological sample (*Homo sapiens*) | Human red blood cells | UK NHS Blood and Transplant; University Medical Center Hamburg- Eppendorf (UKE), Germany | | Provided anonymised. |
| Antibody | 3 F10 High affinity anti-HA (rat monoclonal) | Roche | Cat# 11867423001, RRID:AB_390918 | IFA (1:500), western blot (1:1000) |
| Antibody | Biotinylated anti-rat (goat polyclonal) | Sigma-Aldrich | Cat# AP183B, RRID:AB_92595 | IFA (1:1000), western blot (1:8000) |
| Antibody | anti-aldolase (rabbit polyclonal) | **Mesén-Ramírez et al., 2016** | | IFA (1:2000) |
| Antibody | goat-anti-rat-800CW (goat polyclonal) | LI-COR Biosciences | Cat# 925–32219, RRID:AB_2721932 | Western blot (1:10,000) |
| Antibody | goat-anti-rabbit-680RD (goat polyclonal) | LI-COR Biosciences | Cat# 925–68071, RRID:AB_2721181 | Western blot (1:10,000) |
| Antibody | Goat anti-rat-AlexaFluor 594 (goat polyclonal) | ThermoFisher | Cat# A-11007, RRID:AB_10561522 | IFA (1:2000) |
| Chemical compound, drug | AlexaFluor 594 conjugated Streptavidin | ThermoFisher | Cat# S32356 | |
| Chemical compound, drug | Streptavidin peroxidase | Sigma-Aldrich | Cat# S2438 | |
| Chemical compound, drug | WR99210 | Sigma-Aldrich | Cat# W1770 | |
| Chemical compound, drug | Rapamycin | Sigma-Aldrich | Cat# R0395-1MG | |
| Chemical compound, drug | Compound 2 (4-[7-dimethylamino)methyl]−2-(4-fluorphenyl)imidazo[1,2-α]pyridine-[3-yl]pyrimidin-2-amine | LifeArc (https://www.lifearc.org/) | | Kindly provided by Dr. Simon A. Osborne (LifeArc). |
| Chemical compound, drug | SYBR Green I | ThermoFisher | Cat# S7563 | |
| Chemical compound, drug | rapalog (AP21967) | Clontech | Cat# 635057 | |

*Continued on next page*

*Continued*

| Reagent type (species) or resource | Designation | Source or reference | Identifiers | Additional information |
|---|---|---|---|---|
| Chemical compound, drug | Neomycin/G418 | Sigma-Aldrich | Cat# G418-RO | 400 µg/ml |
| Chemical compound, drug | blasticidin S HCl | Invitrogen | Cat# R21001 | 2 µg/ml |
| Chemical compound, drug | DSM1 | BEI Resources | | 0.9 µM |
| Chemical compound, drug | choline chloride | Sigma-Aldrich | Cat# C7017 | 1 mM |
| Chemical compound, drug | ethanolamine | Sigma-Aldrich | Cat# E9508 | 100 µM |
| Chemical compound, drug | L-serine | Sigma-Aldrich | Cat# S4500 | 2 mM |
| Chemical compound, drug | sn-glycero-3-phosphocholine | Cayman chemical | Cat# 20736 | 1 mM |
| Chemical compound, drug | $^2$H choline-labelled lysoPC | *Brancucci et al., 2017* | | 20 µM. Kindly provided by Dr. Matthias Marti. |
| Chemical compound, drug | DGTS 32:0 | Avanti Polar Lipids | Cat# 857464 | |
| Commercial assay, kit | Ligation Sequencing Kit | Oxford Nanopore Technologies | Cat# SQK-LSK109 | |
| Commercial assay, kit | Native Barcoding Expansion 1–12 | Oxford Nanopore Technologies | Cat# EXP-NBD104 | |
| Commercial assay, kit | MinION flow cell | Oxford Nanopore Technologies | Cat# R9.4.1 | |
| Software, algorithm | BD FACSDiva software | BD Bioscience | RRID:SCR_001456 | |
| Software, algorithm | FlowJo for Mac (version 10.3.0) software | Becton Dickinson Life Sciences | RRID:SCR_008520 | |
| Software, algorithm | Fiji (Image J version 2.0) software | Imagej.net | RRID:SCR_003070 | |
| Software, algorithm | Thermo Xcalibur v3.0.63 software | Thermo Scientific | RRID:SCR_014593 | |
| Software, algorithm | Free Style v1.5 | Thermo Scientific | RRID:SCR_022877 | |
| Software, algorithm | Progenesis QI | Nonlinear Dynamics | RRID:SCR_018923 | |
| Software, algorithm | LipidMatch | *Koelmel et al., 2017* | | |
| Software, algorithm | TraceFinder v5.1 | Thermo Scientific | | |
| Software, algorithm | MS-Dial v4.80 | *Tsugawa et al., 2015* | | |
| Software, algorithm | MinKNOW v20.10 | Oxford Nanopore Technologies; https://community.nanoporetech.com/downloads | | |
| Software, algorithm | Guppy v3.2.2 | Oxford Nanopore Technologies; https://community.nanoporetech.com/downloads | | |
| Software, algorithm | IGV v2.9.4 | *Robinson et al., 2011*; https://software.broadinstitute.org/software/igv/ | | |

*Continued on next page*

*Continued*

| Reagent type (species) or resource | Designation | Source or reference | Identifiers | Additional information |
|---|---|---|---|---|
| Software, algorithm | PDBeFold server | https://www.ebi.ac.uk/msd-srv/ssm/ | | |
| Software, algorithm | ICM-Pro v3.9–1 c/MacOSX | Molsoft LLC; https://www.molsoft.com/icm_pro.html | | |
| Software, algorithm | Muscle v3.8.31 | *Edgar, 2004* | RRID:SCR_011812 | |
| Software, algorithm | trimAl v1.2 | *Capella-Gutiérrez et al., 2009* | RRID:SCR_017334 | |
| Software, algorithm | RAxML | *Stamatakis, 2014* | RRID:SCR_006086 | |
| Software, algorithm | the iTOL server | *Letunic and Bork, 2021* | RRID:SCR_018174 | |
| Software, algorithm | InterProScan | *Jones et al., 2014* | RRID:SCR_006695 | |
| Software, algorithm | myDomains | *Hulo et al., 2008* | | |
| Software, algorithm | R v4.0.2 | http://www.r-project.org/ | RRID:SCR_001905 | |
| Software, algorithm | ggplot2 | https://cran.r-project.org/web/packages/ggplot2/index.html | RRID:SCR_014601 | |
| Software, algorithm | RStudio | http://www.rstudio.com/ | RRID:SCR_000432 | |
| Other | Pierce Anti-HA Magnetic Beads | Thermo Scientific | Cat# 88836 | Magnetic beads conjugated with highly specific anti-HA monoclonal antibody (clone 2–2.2.14). For immunoprecipiation of HA-tagged proteins. |
| Other | AMPure XP beads | Beckman Coulter | Cat# A63881 | Paramagnetic beads that selectively binds to DNA of length greater than 100 bp. Used for high recovery/purification of genomic or amplicons from other contaminants. |

## Plasmid construction

Modification plasmids to produce the four modified *P. falciparum* lines used in this study were constructed as follows.

Targeted gene disruption (TGD) of PfGDPD was attempted using the TGD construct pSLI-PF3D7_1406300-TGD. To generate this, the N-terminal 360 bp of the *gdpd* gene was amplified by PCR using primers PF3D7_1406300-TGD-fw/ PF3D7_1406300-TGD-rev and cloned into pSLI-TGD (*Birnbaum et al., 2017*) using NotI/MluI.

The GDPD:GFP line was made by endogenously tagging PfGDPD with GFP using construct pSLI-PF3D7_1406300-GFP-GlmS-WT, which was generated by amplifying the C-terminal 858 bp of the endogenous *gdpd* gene (PF3D7_1406300) by PCR using primers PF3D7_1406300-TAG-fw/ PF3D7_1406300-TAG-rev and cloning the PCR product into pSLI-GFP-GlmS-WT (*Burda et al., 2020*) using NotI/MluI.

The conditional knockout GDPD:loxPint:HA line was produced by modifying the endogenous gdpd locus in the DiCre-expressing *P. falciparum* B11 line using Cas9-mediated genome editing (*Ghorbal et al., 2014*). A two-plasmid system was used where a targeting plasmid delivers Cas9 and guide RNA to target the PfGDPD locus while a repair plasmid delivers the repair template for homology-directed repair of the Cas9-nicked locus. Two RNA targeting sequences (CATCAATCGTTGGTCATAGA and ACGGAGTAGAATTGGACGTA) were inserted into the pDC2 Cas9/gRNA/hDHFR (human dihydro-folate reductase)/yFCU (yeast cytosine deaminase/uridyl phosphoribosyl transferase)-containing plasmid as described previously (*Knuepfer et al., 2017*) to generate two different targeting plas-mids (pCas9_1406300_gRNA01 and pCas9_1406300_gRNA02 respectively). For the repair plasmid, a 1666 bp long synthetic DNA fragment containing a recodonised segment of almost the complete PfGDPD gene (69–1425 bp; 24–475 aa) flanked upstream by a loxPint module (*Jones et al., 2016*) and downstream by a triple hemagglutinin (HA) tag, a stop codon and a loxP site, was synthesized

commercially (GeneArt, Thermo Fisher Scientific). A 711 bp long 5′ homology arm was amplified from parasite genomic DNA (using primers gdpd_5hom.F/gdpd_5hom.R) and inserted into the synthesized plasmid using NotI/AvrII restriction/ligation. Similarly, a 665 bp long 3′ homology arm was amplified (using primers gdpd_3hom.F/gdpd_3hom.R) and inserted using NheI/XhoI restriction/ligation reaction to produce the final repair plasmid, pREP-GDPD. The repair plasmid was linearised with AgeI overnight prior to transfection.

The conditional knockout line, GDPD:loxPint:HA:Neo-R, was produced by modifying the endogenous gdpd locus using plasmid pSLI-PF3D7_1406300-loxPint:HA:T2A:Neo. To generate this, a DNA fragment consisting of a 458 bp N-terminal targeting sequence, a loxPint module and a recodonized and 3xHA-tagged PfGDPD C-terminal coding sequence was synthesized commercially (GeneScript) and cloned into the pSLI-loxPint:HA:T2A:Neo plasmid (*Davies et al., 2020*) using BglII/SalI.

Gene complementation vectors were constructed by amplifying the PfGDPD coding sequence without stop codon using primers PF3D7_1406300-COMP-fw /PF3D7_1406300-COMP-rev and cloning the PCR product via XhoI/SpeI into the pNMD3:1xNLS-FRB-mCherry-DHODH plasmid (*Birnbaum et al., 2017*), thus replacing the 1xNLS-FRB sequence with the PfGDPD coding sequence to obtain pNMD3:PF3D7_1406300-mCherry-DHODH. Mutagenesis of the putative active site residues (H29, H78) and a putative metal-binding residue (E283) to alanine was performed by overlap extension PCR.

CloneAmp HiFi PCR Premix (TakaraBio) and Phusion High-Fidelity DNA polymerase (New England BioLabs) were used for PCR reactions for all plasmid constructions. All plasmid sequences were confirmed by Sanger sequencing.

For sequences of oligonucleotides and other synthetic DNA used in this study, please refer to *Supplementary file 1*.

## Parasite culture maintenance, synchronisation, and transfection

The DiCre-expressing *P. falciparum* B11 line (*Perrin et al., 2018*) was maintained at 37°C in human RBCs in RPMI 1640 containing Albumax II (Thermo Fisher Scientific) supplemented with 2mM L-glutamine. Synchronisation of parasite cultures were done as described previously (*Harris et al., 2005*) by isolating mature schizonts by centrifugation over 70% (v/v) isotonic Percoll (GE Healthcare, Life Sciences) cushions, letting them rupture and invade fresh erythrocytes for 2 hr at 100 rpm, followed by removal of residual schizonts by another Percoll separation and sorbitol treatment to finally obtain a highly synchronised preparation of newly invaded ring-stage parasites. To obtain the GDPD:HA:loxPint line, transfections were performed by introducing DNA into ~$10^8$ Percoll-enriched schizonts by electroporation using an Amaxa 4D Nucleofector X (Lonza), using program FP158 as previously described (*Moon et al., 2013*). For Cas9-based genetic modifications, 20 μg of targeting plasmid and 60 μg of linearised repair template were electroporated. Drug selection with 2.5 nM WR99210 was applied 24 hr post-transfection for 4 days with successfully transfected parasites arising at 14–16 days. Clonal transgenic lines were obtained by serial limiting dilution in flat-bottomed 96-well plates (*Thomas et al., 2016*) followed by propagating wells that contain single plaques. Successful integration was confirmed by running diagnostic PCR either directly on culture using BloodDirect Phusion PCR premix or from extracted genomic DNA (DNAeasy Blood and Tissue kit, Qiagen) with CloneAmp HiFi PCR Premix (TakaraBio).

To obtain the GDPD:loxPint:HA$_{NF54}$ line, the same procedure as detailed above was followed with the DiCre-expressing *P. falciparum* NF54 line (*Tibúrcio et al., 2019*).

*P. falciparum* 3D7 line was maintained at 37°C in an atmosphere of 1% $O_2$, 5% $CO_2$, and 94% $N_2$ and cultured using RPMI complete medium containing 0.5% Albumax according to standard procedures (*Trager and Jensen, 1976*). For generation of stable integrant cell lines, GDPD:GFP and GDPD:loxPint:HA:Neo-R, mature schizonts of 3D7 parasites were electroporated with 50 μg of plasmid DNA using a Lonza Nucleofector II device (*Moon et al., 2013*). Parasites were first selected in medium supplemented with 3 nM WR99210 (Jacobus Pharmaceuticals). Parasites containing the episomal plasmids selected with WR99210 were subsequently grown with 400 μg/ml Neomycin/ G418 (Sigma) to select for integrants carrying the desired genomic modification as described previously (*Birnbaum et al., 2017*). Successful integration was confirmed by diagnostic PCR using FIREpol DNA polymerase (Solis BioDyne). Transgenic GDPD:loxPint:HA:Neo-R parasites were then further transfected with the plasmid pSkipFlox (*Birnbaum et al., 2017*) and selected with 2 μg/

ml blasticidin S (Invitrogen) for constitutive expression of the DiCre recombinase under the *crt* promoter. For co-expression of a PV marker, GDPD:GFP parasites were further transfected with a plasmid expressing a signal peptide conjugated with the mScarlet coding sequence (SP-mScarlet) under the constitutive *nmd3* promoter (*Mesén-Ramírez et al., 2019*). For gene complementation, GDPD:loxPint:HA:Neo-R parasites were further transfected with wildtype or mutated versions of the pNMD3:PF3D7_1406300-mCherry-DHODH plasmid and transfectant parasites were selected for with 0.9 μM DSM1 (BEI Resources).

To obtain GDPD-null parasites, DiCre-mediated excision of target locus was induced by rapamycin treatment (100 nM RAP for 3 hr or 10 nM overnight) of synchronous early ring-stage parasites (2–3 hr post-invasion) as previously described (*Collins et al., 2013*). DMSO treated parasites were used as wildtype controls.

To induce sexual differentiation, GDPD:loxPint:HA$_{NF54}$ cultures were treated with either conditioned media (-RAP+CM) to induce sexual commitment, DMSO (-RAP) or rapamycin (+RAP) to induce PfGDPD gene knockout. Cultures were fed daily and diluted when asexual stages reached high parasitaemia.

For GDPD:loxPint:HA:Neo-R parasites, 250 nM rapalog (AP21967, Clontech, stored at -20°C as a 500 mM stock in ethanol; working stocks were kept as 1:20 dilutions in RPMI at 4°C) was used to induce excision. Medium was changed daily and fresh rapalog was added every day.

## Western blot and immunofluorescence assays

Ablation of PfGDPD was assessed by western blotting and immunofluorescence-based detection of the triple HA tagged GDPD. For GDPD:loxPint:HA parasites, proteins were extracted from 24 hr trophozoites (after saponin lysis) or 45 hr schizonts directly into SDS buffer and resolved by SDS polyacrylamide gel electrophoresis (SDS-PAGE) and transferred to nitrocellulose membrane (Supported nitrocellulose membrane, Bio-Rad). Membranes were blocked with 5% bovine serum albumin (BSA) in PBS-T (0.05% Tween 20) and subsequently probed with the rat anti-HA mAb 3F10 (Sigma, 1:1000 dilution), followed by biotin-conjugated anti-rat antibody (Roche, 1:8000) and then with horseradish peroxidase-conjugated streptavidin (Sigma, 1:10,000). Immobilon Western Chemiluminescent HRP Substrate (Millipore) was used according to the manufacturer's instructions, and blots were visualized and documented using a ChemiDoc Imager (Bio-Rad) with Image Lab software (Bio-Rad). Antibodies against HSP70 (a gift from E. Knuepfer, Francis Crick Institute) was used at 1:2000 as loading control. For Coomassie Blue staining, SDS-PAGE gels were stained with InstantBlue Coomassie Protein Stain (Abcam) for half an hour and destained with water overnight.

For GDPD:loxPint:HA:Neo-R parasites, protein samples were resolved by SDS-PAGE and transferred to nitrocellulose membranes (LICOR). Membranes were blocked in 5% milk in TBS-T followed by incubation in the following primary antibodies that were diluted in TBS-T containing 5% milk: rat-anti-HA 3F10 (Sigma, 1:1000) and rabbit-anti-aldolase (1:2000) (*Mesén-Ramírez et al., 2016*) antibodies. Subsequently, membranes were incubated in similarly diluted secondary antibodies: goat-anti-rat-800CW (LICOR, 1:10000) and goat-anti-rabbit-680RD (LICOR, 1:10000) and scanned on a LICOR Odyssey FC imager.

For immunofluorescene assays of GDPD:loxPint:HA parasites, thin films of parasite cultures containing C2-arrested mature schizonts were air-dried, fixed in 4% (w/v) formaldehyde for 30 min (Agar Scientific Ltd.), permeabilized for 10 min in 0.1% (w/v) Triton X-100 and blocked overnight in 3% (w/v) bovine serum albumin in PBS. Slides were probed with rat anti-HA mAb 3F10 (1:500 dilution) to detect GDPD-3HA. Primary antibodies were detected by probing with biotin-conjugated anti-rat antibody (1:1,000) followed by Alexa 594-conjugated streptavidin (Invitrogen, 1:1000). Slides were then stained with 1 μg/mL DAPI, mounted in Citifluor (Citifluor Ltd., Canterbury, U.K.).

For GDPD:loxPint:HA:Neo-R parasites, IFA was performed in solution. Parasites were fixed with 4% paraformaldehyde / 0.0075% glutaraldehyde in PBS for 10 min at RT, permeabilized in 0.1% Triton X-100 in PBS for 5 min and blocked for 10 min in 3% BSA/PBS. Samples were probed with rat anti-HA 3F10 (Sigma, 1:1,000) in blocking buffer. Bound primary antibodies were detected using goat-anti-rat-AlexaFluor594 secondary antibodies (Thermo Scientific) diluted 1:2000 in blocking buffer additionally containing 1 μg/mL DAPI for visualization of nuclei.

## Fluorescence microscopy

For live cell microscopy of GDPD:GFP parasites, parasites were incubated with 1 µg/mL DAPI in culture medium for 15 min at 37°C to stain nuclei before microscopic analysis. GDPD:loxPint:HA parasites were imaged using AxioVision 3.1 software on an Axioplan 2 Imaging system (Zeiss) wtih a Plan-APOCHROMAT 100×/1.4 oil immersion objective. All other parasites lines were imaged on a Leica D6B fluorescence microscope, equipped with a Leica DFC9000 GT camera and a Leica Plan Apochromat 100 x/1.4 oil objective. Image processing was performed using ImageJ (*Schneider et al., 2012*).

## Growth and replication assays

For GDPD:loxPint:HA parasites, growth assays were performed to assess parasite growth across 3–4 erythrocytic replication cycles. Synchronous cultures of ring-stage parasites at 0.1% parasitaemia and 2% haematocrit were maintained in triplicates in 12-well plates. To assess if exogenous precursors can rescue the growth defect in GDPD-null parasites, cultures were grown in the presence or absence of 1 mM choline chloride (Sigma), 1 mM glycerophosphocholine (Sigma), 100 µM ethanolamine (Sigma) or 2 mM serine (Sigma). Fresh precursor-supplemented media was provided at around 24 hpi of each erythrocytic cycle. To assess the effect of choline on GDPD-null parasites (G1 parasite line), cultures (0.1% parasitaemia) were grown in the presence, absence or titrated concentrations of choline chloride for two cycles and final parasitaemia was estimated.

50 µL from each well was sampled at 0, 2, 4, and 6 days post-RAP treatment, fixed with 50 µL of 0.2% glutaraldehyde in PBS and stored at 4°C for flow cytometry quantification. Fixed parasites were stained with SYBR Green (Thermo Fisher Scientific, 1:10,000 dilution) for 20 min at 37°C and analysed by flow cytometry on a BD FACSVerse using BD FACSuite software. For every sample, parasitaemia was estimated by recording 10,000 events and filtering with appropriate forward and side scatter parameters and gating for SYBR Green stain-positive (infected RBCs) and negative RBCs using a 527/32 detector configuration. All data were analysed using FlowJo software. Average fold increase in parasitaemia was calculated by averaging fold increase in parasitaemia between cycle 1, 2, and 3.

Growth stage progression was monitored by microscopic examination at selected timepoints using Giemsa-stained thin blood films. Samples were also fixed at these timepoints for flow cytometry analysis. Fluorescence intensity of the SYBR Green stain-positive population was quantified to assess DNA content, the increase of which was taken as a proxy for growth stage progression.

Merozoite numbers were estimated from Giemsa-stained blood films of schizonts let to mature by arresting egress using C2 (1 µM) for 3 hr.

To assess invasion rates, highly synchronous mature schizonts were added to fresh erythrocytes (2% haematocrit) and let to invade for 4 hr at both static and mechanical shaking (100 rpm) conditions (four replicates in each condition). Cultures were sampled before and after the 4 hr invasion and fixed as before for quantification.

For growth analysis of GDPD:loxPint:HA:Neo-R parasites, parasitaemia was analyzed by flow cytometry at 1, 3, 5, and 7 days after Rapa addition, when most of the parasites were at the trophozoite stage. parasitemia was analyzed at 1, 3, 5, and 7 days post Rapalog addition. Cultures were diluted 10-fold into fresh RBCs after the 5th day to prevent overgrowth. Parasitaemia assessment was performed as described previously (*Malleret et al., 2011*). In brief, 20 µL resuspended parasite culture was incubated with dihydroethidium (5µg/ml, Cayman) and SYBR Green I dye (0.25 x dilution, Invitrogen) in a final volume of 100 µL medium for 20 min at RT protected from light. Samples were analyzed (100,000 events) on a ACEA NovoCyte flow cytometer. For quantification of developmental stages, synchronous ring stage cultures were diluted to ~0.1 and ~1% parasitaemia and Giemsa-stained blood films were prepared at 40/48 hpi (1% starting parasitemia) and 88/96 hpi (0.1% starting parasitemia). For stage quantification, at least 20 fields of view were recorded using a 63 x objective per sample. Erythrocyte numbers were then determined using the automated Parasitaemia software (http://www.gburri.org/parasitemia/) and the number of the different parasite stages was manually counted on these images.

## Transmission electron microscopy

GDPD:loxPint:HA synchronous ring stage parasites were treated with RAP and allowed to progress to 40 hr in the next cycle to obtain a population of growth arrested GDPD-null parasites. These parasites

were fixed with 2.5% glutaraldehyde/ 4% formaldehyde in 0.1 M phosphate buffer (PB) for 30 min at room temperature (RT). Parasites were embedded in 3% low melting point agarose and cut into 1 mm$^3$ blocks. The blocks were then processed using a modified version of the NCMIR protocol (**Deerinck et al., 2010**). Briefly, blocks were washed in 0.1 M PB and post-fixed with 1% reduced osmium (1% OsO$_4$ /1.5% K$_3$Fe(CN)$_6$) for 1 hr at 4°C, and then washed in double distilled water (ddH$_2$O). The blocks were incubated in 1% thiocarbohydrazide (TCH) for 20 min at RT, rinsed in ddH$_2$O and further fixed with 2% osmium tetroxide for 30 min at RT. The blocks were then stained with 1% uranyl acetate at 4°C overnight, washed in ddH$_2$O and stained with Walton's lead aspartate for 30 min at 60°C. The blocks were washed in ddH$_2$O and dehydrated stepwise using serial dilutions of ethanol: 30% and 50% at RT for 5 min each, then 70%, 90% and 2 x 100% for 10 min each. The blocks were infiltrated with 4:1 mixture of propylene oxide (PO):Durcupan resin (Sigma Aldrich) for 1 hr at RT, followed by 1:1 and 1:4 mixtures for 1 hr each at RT, then with 100% Durcupan resin for 48 hr. Blocks were polymerised in fresh Durcupan resin at 60°C for 48 hr. The samples were cut into 70 nm ultrathin sections using an ultramicrotome (UC7, Leica Microsystems UK) and picked up onto copper mesh grids (Agar Scientific). Images were obtained on a 120 kV transmission electron microscope (TEM) (Tecnai G2 Spirit BioTwin, Thermo Fisher Scientific) using a charge-coupled device camera (Oneview, Gatan Inc).

## Polarized light microscopy

Haemozoin content was visualized and quantified in methanol-fixed thin blood films using transmitted polarized light (488 nm) in a Zeiss Axio Observer.Z1 microscope fitted with a 63 x/1.4NA Plan Apochromat objective, transmitted white light LED (Thorlabs) and imaged with a Hamamatsu OrcaSpark CMOS camera. A polarizer was placed above the sample and an analyser module in the filter turret below the sample. The polarizer was rotated to cross with the analyser at 90°. Only well-focussed haemozoin signals were chosen for quantification using FIJI and around 50 parasites were measured for each group and timepoint.

## Lipidomic profiling and metabolic labeling assays

To assess the changes in phospholipid content due to absence of GDPD, total phospholipids from GDPD-null and wildtype schizonts were extracted and lipid species were determined and quantified by LC-MS/MS.

Schizonts were isolated using Percoll cushions from RAP- and DMSO-treated GDPD:loxPint:HA parasitized cultures (100 ml, 0.5% haematocrit, 35–40% parasitaemia) grown for 45 hr post treatment and allowed to mature for 4 hr at 37°C in the presence of egress-blocking C2 (1 μM) in order to achieve a high level of homogeneity in the samples. Egress-blocked schizonts were washed twice with RPMI media without Albumax II (with C2 at 1 μM) and subject to lipid extraction. Experiments were carried out in triplicates.

For metabolic labeling experiments, RAP- and DMSO-treated GDPD:loxPint:HA parasitized cultures (10 mL, 1% haematocrit, 10% parasitaemia) were grown for 14 hr (from 28±1 hpi to 42±1 hpi) either in the presence (four replicates) or absence (one replicate) of 20 μM $^2$H choline-labelled lysoPC (a kind gift from Dr Matthias Marti; **Brancucci et al., 2017**). For labeling experiments comparing GDPD-null clonal parasite line G1 with GDPD:loxPint:HA (B4 line), both parasite lines were maintained for one cycle in the presence of 1 mM choline. Choline was removed at the start of the next cycle and choline-deprived parasites were maintained for a further 18 hr (from 24±1 hpi to 42±1 hpi) either in the presence (three replicates) or absence (one replicate) of 20 μM $^2$H choline-labelled lysoPC. At 42 hpi, cultures were spun down (3600 rpm, 3 min) and RBC pellets were lysed with 0.015% ice-cold saponin for 10 min on ice following which parasites were spun down (6000 rpm, 3 min) and washed five times with ice-cold PBS. Saponin lysis and washes were repeated in the case of incomplete lysis in some samples. Parasite pellets were resuspended in ice-cold PBS and subjected to lipid extraction procedures.

Lipid extraction for each sample was performed by adding 400 μL of approximately 1x10$^8$ parasites to each of three tubes (technical replicates) that contained 600 μL methanol and 200 μL chloroform. Samples were sonicated for 8 min at 4°C and incubated at 4°C for 1 hr. 400 μL of ice-cold water was added (thus obtaining the 3:3:1 water:methanol:chloroform ratio) to the samples, mixed well and centrifuged at max speed for 5 min at 4°C for biphasic partitioning. The lower apolar phase was added to fresh tubes. The upper aqueous layer was removed and lipids were extracted once more

by adding 200 µL of chloroform, vortexing and centrifuging as before. The apolar phases from both extractions were pooled (400 µL) and dried under nitrogen stream and resuspended in butanol/methanol (1:1,v/v) containing 5 µM ammonium formate.

## Affinity purification and in vitro enzymatic assay

To determine whether PfGDPD can break down GPC and release choline, PfGDPD-HA was affinity purified from parasite protein lysates and treated with GPC substrate in vitro. Around 50 µL of frozen schizont pellets of different parasite lines were lysed with 1 mL of 0.15% saponin for 20 min on ice followed by centrifugation at 13,000 rpm at 4°C for 10 min. 950 µL of the saponin lysate was incubated with 50 µL of washed anti-HA magnetic beads (Pierce) in rotary mixer for 2 hr at 4°C. PfGDPD-HA bound beads were magnet separated and washed thrice with 3001 ice-cold TBS-T (Tris buffered saline with 0.01% Tween-20) and then four times with 300 µL ice-cold reaction buffer (100 mM HEPES pH 7.5, 150 mM NaCl, 10mM $MgCl_2$) to remove traces of Tween-20. Aliquots of the beads (resuspended in reaction buffer) were treated with 50 µL of 1 mM GPC and incubated at 37°C for various durations (5, 10, 15, and 240 min) with regular mixing. Mock reactions without beads or GPC were set up to account for any spontaneous breakdown of GPC or any GPC/choline carryover from the lysate respectively. Reactions were stopped by placing the tubes on ice, beads were magnet-separated and supernatants were stored at -20°C, to be analysed by LC-MS/MS.

## MS/MS run and subsequent analysis

For whole cell lipidomic analysis, the LC-MS method was adapted from *Greenwood et al., 2019*. Cellular lipids were separated by injecting 10 µL aliquots onto a column: 2.1×100 mm, 1.8 µM C18 Zorbax Elipse plus column (Agilent) using an Dionex UltiMate 3000 LC system (Thermo Scientific). A 20 min elution gradient of 45% to 100% Solvent B was used, followed by a 5 min wash of 100% Solvent B and 3 min re-equilibration, where Solvent B was water:acetonitrile:isopropanol, 5:20:75 (v/v/v) with 10mM ammonium formate (Optima HPLC grade, Fisher Chemical) and Solvent A was 10 mM ammonium formate in water (Optima HPLC grade, Fisher Chemical). Other parameters were as follows: flow rate 600 µL/min; column temperature 60°C; autosampler temperature 10°C. MS was performed with positive/negative polarity switching using an Q Exactive Orbitrap (Thermo Scientific) with a HESI II probe. MS parameters were as follows: spray voltage 3.5 kV and 2.5 kV for positive and negative modes, respectively; probe temperature 275°C; sheath and auxiliary gases were 55 and 15 arbitrary units, respectively; full scan range: 150–2000 m/z with settings of auto gain control (AGC) target and resolution as Balanced and High ($3 \times 10^6$ and 70,000), respectively. Data was recorded using Xcalibur 3.0.63 software (Thermo Scientific). Mass calibration was performed for both ESI polarities before analysis using the standard Thermo Scientific Calmix solution. To enhance calibration stability, lock-mass correction was also applied to each analytical run using ubiquitous low-mass contaminants. To confirm the identification of significant features, pooled quality control samples were ran in data-dependent top-N (ddMS2-topN) mode, acquisition parameters as follows: resolution of 17,500, auto gain control target under $2 \times 10^5$, isolation window of m/z 0.4 and stepped collision energy 10, 20 and 30 in HCD (high-energy collisional dissociation) mode. Qualitative and quantitative analyses were performed using Free Style 1.5 (Thermo Scientific), Progenesis (Nonlinear Dynamics), and LipidMatch (*Koelmel et al., 2017*).

For metabolic labelling experiments, LC-MS was performed as above. Qualitative and quantitative analyses were performed using Free Style 1.5 and TraceFinder 5.1, respectively (both Thermo Scientific). Label incorporation was calculated by comparison of unlabelled lysoPC ions to their labelled (M+9 isotopologue) counterpart. LipidMatch was used for identification confirmation (*Koelmel et al., 2017*).

For DGTS identification, the LC-MS method was adapted from a previously published method (*Greenwood et al., 2019*). Briefly, lipids were separated using a 2.1x100 mm, 1.9 µM C18 Zorbax Elipse plus column (Agilent) using a Dionex UltiMate 3000 LC system (Thermo Scientific). Analytes were separated using 10mM ammonium formate in water (Optima HPLC grade, Sigma Aldrich) as solvent A and water:acetonitrile:isopropanol, 5:20:75 (v/v/v) with 10 mM ammonium formate (Optima HPLC grade, Sigma Aldrich) as solvent B at 0.6 mL/min flow rate. A 20 min elution gradient of 45% to 100% Solvent B was used, followed by a 5 min wash of 100% Solvent B and 3 min re-equilibration.

Other parameters were as follows: column temperature 60°C ; injection volume 5 μL; autosampler temperature 10°C.

MS was performed with positive/negative polarity switching using a Q Exactive Orbitrap (Thermo Scientific) with a HESI II probe. MS parameters were as follows: spray voltage 3.5 kV and 2.5 kV for positive and negative modes, respectively; probe temperature 275°C ; sheath and auxiliary gases were 55 and 15 arbitrary units, respectively; full scan range: 150–2000 m/z with settings of AGC target and resolution as Balanced and High ($3 \times 10^6$ and 70,000) respectively. Data was recorded using Xcalibur 3.0.63 software (Thermo Scientific). Mass calibration was performed for both ESI polarities before analysis using the standard Thermo Scientific Calmix solution. To enhance calibration stability, lock-mass correction was also applied to each analytical run using ubiquitous low-mass contaminants. To confirm the identification of significant features, pooled quality control samples and DGTS 32:0 (Avanti Polar Lipids) were run in Parallel Reaction Monitoring (PRM) mode, with acquisition parameters as follows: auto gain control target under 2, isolation window of m/z 0.4, stepped collision energy 35, 40 and 45 in HCD mode and resolution of 35,000. For PRM, the included ions are listed in *Figure 5C*; -sourcedata3. Qualitative analysis was performed using Xcalibur FreeStyle 1.8 SP1 software (Thermo Scientific) according to the manufacturer's workflows and MSDial 4.80.

For analysing GPC and choline content, data acquisition was performed using an adaptation of a method previously described (*Creek et al., 2011*). The supernatant after enzymatic reaction was diluted (1:300) in methanol:water (1:1) and injected into a Dionex UltiMate 3000 LC system (Thermo Scientific) with a ZIC-pHILIC (150 mm x 4.6 mm, 5 μM particle) column (Merck Sequant). Analytes were separated using 20mM ammonium carbonate in water (Optima HPLC grade, Sigma Aldrich) as solvent A and acetonitrile (Optima HPLC grade, Sigma Aldrich) as solvent B at 0.3 mL/min flow rate. The elution started at 80% solvent B, maintained for 2 min, followed by 15 min elution gradient of 80% to 5% solvent B, with 3 min wash of 5% solvent B and 5 min re-equilibration to 80% solvent B. Other parameters were as follows: column temperature 25°C; injection volume 10 μL ; autosampler temperature 4°C.

MS was performed with positive/negative polarity switching using an Q Exactive Orbitrap (Thermo Scientific) with a HESI II (Heated electrospray ionization) probe. MS parameters were as follows: spray voltage 3.5 kV and 3.2 kV for positive and negative modes, respectively; probe temperature 320°C; sheath and auxiliary gases were 30 and 5 arbitrary units, respectively; full scan range: 50–750 m/z with settings of AGC (auto gain control target) and resolution as Balanced and High ($3 \times 10^6$ and 70,000), respectively. Data were recorded using Xcalibur 3.0.63 software (Thermo Scientific). Mass calibration was performed for both ESI polarities before analysis using the standard Thermo Scientific Calmix solution. To enhance calibration stability, lock-mass correction was also applied to each analytical run using ubiquitous low-mass contaminants. Full MS/dd-MS2 (Top N) acquisition parameters for metabolite identification using pooled quality control samples (PBQC), and choline and GPC standard mix (1 μM): resolution 17,500; collision energies stepped collision energy 10, 20 and 30 in HCD (high-energy collisional dissociation) mode, with choline ($[M]^+$ and $[M+H]^+$) and GPC ($[M+H]^+$) mass inclusion. Qualitative analysis was performed using Xcalibur FreeStyle 1.8 SP1 software (Thermo Scientific) according to the manufacturer's workflows and MSDial 4.80.

## DNA sequencing

To determine the proportion of excised and unexcised parasites in the population that emerged upon choline supplementation, we sequenced their genomes using Nanopore sequencing (Oxford Nanopore Technologies). RAP-treated GDPD:loxPint:HA parasites were set up at 0.1% parasitaemia (three replicates) and allowed to grow in the presence of 1mM choline chloride for three erythrocytic cycles. Genomic DNA was extracted from the choline-supplemented parasites and the parent RAP- and DMSO-treated parasites using DNeasy Blood and Tissue kit (Qiagen) and repurified using AMPure beads (Beckman Coulter, 1.8 X sample volume). Five barcoded DNA libraries were prepared using Ligation Sequencing Kit (SQK-LSK109) and Native Barcoding Expansion 1–12 (EXP-NBD104) kits, pooled and sequenced in MinION R9 flow cell according to manufacturer's instructions. Basecalling and demultiplexing was done using Guppy v3.2.2 to yield 200,000–320,000 reads per sample. Reads were mapped onto the Pf3D7 genome using minimap2 v2.2 and mapping visualized in IGV v2.9.4 (*Robinson et al., 2011*).

## Substrate docking in silico

PfGDPD model coordinates were obtained from the AlphaFold Protein Structure database with code AF-Q8IM31-F1 (DeepMind, EMBL-EBI) (*Jumper et al., 2021*). The catalytic region was modeled with very high confidence with a per-residue confidence score above 90%. This model was used to search for similar protein structures within the whole PDB archive using the PDBeFold server (EMBL-EBI, http://www.ebi.ac.uk/msd-srv/ssm) which allows tridimensional alignments of protein structures and list the best Ca-alignments of compared structures. The closely related $Mg^{2+}$ dependent marine phosphodiesterase 4OEC (rmsd = 1.63 Å) was used to place a magnesium ion in the vicinity of the conserved PfGDPD coordinating residues Glu63, Asp65 and Glu283 followed by an energy minimization of the residue side chains using the Internal Coordinate Mechanics software (ICM-Pro) package version 3.9–1 c/MacOSX (Molsoft LLC, San Diego, CA) (*Abagyan et al., 1994*). The glycerol-3-phosphate (G3P)-complexed form of OLEI02445 from *Oleispira antartica* (3QVQ, rmsd = 1.57 Å; not shown) allowed the use of its ligand to define the active site binding pocket of PfGDPD for docking in ICM-Pro.

Non-covalent flexible docking of the phospholipids GroP, GroPCho, GroPSer, and lysoPC(16:0) into the active site of the $Mg^{2+}$ PfGDPD model was performed within ICM-Pro. The lipids were drawn using the ICM chemistry molecular editor to generate a sdf docking table. Hydrogen atoms were added to the $Mg^{2+}$ PfGDPD model and after superimposition of the phosphodiesterase 3QVQ (rmsd = 1.57 Å) in complex with a sn-glycerol-3-phosphate, this ligand was used to define the PfGDPD substrate binding pocket used for the docking procedure. The potential energy maps of the PfGDPD receptor pocket and docking preferences were set up using the program default parameters. Energy terms were based on the all-atom vacuum force field ECEPP/3 and conformational sampling was based on the biased probability Monte Carlo (BPMC) procedure (*Abagyan and Totrov, 1994*). Three independent docking runs were performed per ligand, with a length of simulation (thoroughness) varied from 3 to 5 and the selection of 2 docking poses. Ligands were ranked according to their ICM energetics (ICM score, unitless), which weighs the internal force-field energy of the ligand combined with other ligand-receptor energy parameters.

## Phylogenetic analysis

Orthologs of PfGDPD in other apicomplexan parasites and ancestors were identified and their sequences retrieved from OrthoMCL DB Release 6.4 (https://orthomcl.org/orthomcl/app) and VEuPathDB Release 51 (https://veupathdb.org/veupathdb/app). Multiple sequence alignment was performed using Muscle v3.8.31 (*Edgar, 2004*) with default parameters and the resulting alignment trimmed using trimAl v1.2 (*Capella-Gutiérrez et al., 2009*) with -automated1 setting. A maximum likelihood tree was inferred from the 383 aa long trimmed alignment using RAxML (*Stamatakis, 2014*), using PROTGAMMA model for rate heterogeneity and bootstrapping conducted until the majority-rule consensus tree criterion (-I autoMRE) was satisfied (300 replicates). The resulting phylogenetic tree was visualised in the iTOL server (*Letunic and Bork, 2021*). Protein domains were detected using InterProScan (*Jones et al., 2014*) and visualised using myDomains (*Hulo et al., 2008*).

## Statistical analysis

All statistical analysis and data visualization was performed in R v4.0.2 (*R Development Core Team, 2021*). Student's t-test were used to compare group means and where necessary Bonferroni adjustment for multiple comparisons was applied to the p-value of statistical significance. All statistical analysis are available as R code in https://github.com/a2g1n/GDPDxcute (copy archived at swh:1:rev:eb74f-d96a9eb490579bee938d965406dbbb24769, *Ramaprasad, 2022*).

## Acknowledgements

AR was funded by a Marie Skłodowska Curie Individual Fellowship (Project number 751865). The work was also supported by funding to MJB from the Wellcome Trust (20318/A/20/Z) and the Francis Crick Institute (https://www.crick.ac.uk/) which receives its core funding from Cancer Research UK (CC2129), the UK Medical Research Council (CC2129), and the Wellcome Trust (CC2129). For the purpose of Open Access, the author has applied a CC BY public copyright licence to any Author Accepted Manuscript version arising from this submission. The work was further supported by Wellcome ISSF2 funding

to the London School of Hygiene & Tropical Medicine. PCB acknowledges funding by the German Research Foundation (DFG project number 414222880). We would like to thank Tobias Spielmann for providing the aldolase antibody, and Matthias Marti for providing $^2$H choline-labelled lysoPC. Images were acquired on microscopes of the CSSB imaging facility. We further would like to thank Matt Renshaw and Kurt Anderson (Advanced Light Microscopy, Francis Crick Institute, London) for their help with polarization microscopy.

## Additional information

### Funding

| Funder | Grant reference number | Author |
|---|---|---|
| H2020 Marie Skłodowska-Curie Actions | 751865 | Abhinay Ramaprasad |
| Wellcome Trust | 20318/A/20/Z | Michael J Blackman |
| Cancer Research UK | CC2129 | Abhinay Ramaprasad<br>Enrica Calvani<br>Aaron J Sait<br>Susana Alejandra Palma-Duran<br>Chrislaine Withers-Martinez<br>Fiona Hackett<br>James Macrae<br>Lucy Collinson<br>Michael J Blackman |
| Medical Research Council | CC2129 | Abhinay Ramaprasad<br>Enrica Calvani<br>Aaron J Sait<br>Susana Alejandra Palma-Duran<br>Chrislaine Withers-Martinez<br>Fiona Hackett<br>James Macrae<br>Lucy Collinson<br>Michael J Blackman |
| Wellcome Trust | CC2129 | Abhinay Ramaprasad<br>Enrica Calvani<br>Aaron J Sait<br>Susana Alejandra Palma-Duran<br>Chrislaine Withers-Martinez<br>Fiona Hackett<br>James Macrae<br>Lucy Collinson<br>Michael J Blackman |
| Wellcome Trust | ISSF2 | Michael J Blackman |
| Deutsche Forschungsgemeinschaft | 414222880 | Paul-Christian Burda |

The funders had no role in study design, data collection and interpretation, or the decision to submit the work for publication. For the purpose of Open Access, the authors have applied a CC BY public copyright license to any Author Accepted Manuscript version arising from this submission.

### Author contributions

Abhinay Ramaprasad, Conceptualization, Data curation, Formal analysis, Investigation, Visualization, Methodology, Writing - original draft, Writing - review and editing; Paul-Christian Burda, Conceptualization, Data curation, Formal analysis, Investigation, Methodology, Writing - review and editing;

Enrica Calvani, Susana Alejandra Palma-Duran, Data curation, Formal analysis, Investigation, Methodology; Aaron J Sait, Chrislaine Withers-Martinez, Fiona Hackett, James Macrae, Lucy Collinson, Investigation, Methodology; Tim Wolf Gilberger, Michael J Blackman, Conceptualization, Supervision, Funding acquisition, Writing - review and editing

## Author ORCIDs
Abhinay Ramaprasad (ID) http://orcid.org/0000-0001-9372-5526
Paul-Christian Burda (ID) http://orcid.org/0000-0003-0461-4352
Aaron J Sait (ID) http://orcid.org/0000-0001-6091-0426
James Macrae (ID) http://orcid.org/0000-0002-1464-8583
Tim Wolf Gilberger (ID) http://orcid.org/0000-0002-7965-8272
Michael J Blackman (ID) http://orcid.org/0000-0002-7442-3810

## Decision letter and Author response
Decision letter https://doi.org/10.7554/eLife.82207.sa1
Author response https://doi.org/10.7554/eLife.82207.sa2

## Additional files

### Supplementary files
• MDAR checklist
• Supplementary file 1. Sequences of all oligos and synthesized constructs used in this study.

### Data availability
Sequencing data have been deposited in ENA under Project PRJEB55180. All data generated or analysed are included in the manuscript or provided as source data files. All source codes are available via GitHub (https://github.com/a2g1n/GDPDxcute; copy archived at swh:1:rev:eb74fd96a9eb490579bee938d965406dbbb24769).

The following dataset was generated:

| Author(s) | Year | Dataset title | Dataset URL | Database and Identifier |
|---|---|---|---|---|
| Ramaprasad A | 2022 | A choline-releasing glycerophosphodiesterase essential for phosphatidylcholine biosynthesis and blood stage development in the malaria parasite | https://www.ebi.ac.uk/ena/browser/view/PRJEB55180 | European Nucleotide Archive, PRJEB55180 |

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
