## [Editor Report]

This high-quality study characterizes a key enzyme in asexual red blood stages of the malaria parasites that is used to salvage lipid precursors needed for membrane biogenesis and parasite growth in red blood cells. A previously identified glycerophosphodiesterase (PfGDPD), is shown to mediate the hydrolysis of host lyso-phosphatidycholine to generate choline, which in turn is required for parasite de novo phosphatidylcholine synthesis. Extensive analysis of the localization, growth phenotype and lipidomic profiles of PfGDPD deficient parasites indicate that this salvage pathway is essential for lipid homeostasis and asexual parasite development.

---

## [Decision Letter]

**Decision letter after peer review:**

Thank you for submitting your article "A choline-releasing glycerophosphodiesterase essential for phosphatidylcholine biosynthesis and blood stage development in the malaria parasite" for consideration by *eLife*. Your article has been reviewed by 3 peer reviewers, and the evaluation has been overseen by a Reviewing Editor and Dominique Soldati-Favre as the Senior Editor. The following individual involved in review of your submission has agreed to reveal their identity: Paul A Sigala (Reviewer #2).

Essential revisions:

1) Provide additional panel in Figure 2 confirming localization of PfGDPH with a food vacuole marker

2) Provide further evidence for how exogenous choline rescues GDPD KO parasite lipid levels, ideally via isotope tracing to unambiguously demonstrate utilization of exogenous choline rather than upregulation/utilization of an alternative source.

3) Revise Figure 5 (especially 5C) to clarify the experimental set-up and better match to textual descriptions

4) Discuss or confirm alternative possibilities for role of GDPH in gametocytogenesis

*Reviewer #1 (Recommendations for the authors):*

The gene knock out studies, as well as enzyme activity analysis of PfGDPD were carried out in an earlier study (Denloye et al., 2012), which highlighted its role in parasite survival and possible role in providing choline for phospholipid synthesis in the parasite.

The phospholipid analysis shows a generalized response of disturbance in lipid homeostasis; phospholipid analysis in the first cycle (+RAP/-RAP) shows effect on levels of several phospholipids which are overall down, and a generalized increase in DAGs and TAGs. There is a clear mis-regulation of lipid metabolism in the GDPD-KO conditions, but it seems unclear how exactly it is linked with its proposed functions. The authors highlighted that the parasite can shift from CDP-choline pathway to the SDPM pathway to produce PC, which could be reason for lowered PE and PS production; however, the PS production is independent of PC or PE synthesis, which may suggest generalized effect on phospholipid profiles. Have the authors assessed upregulation of PMT pathway under KO conditions? What is the explanation for DAG and TAG increase under KO conditions?

In the cell biology study of the KO line, parasites show developmental arrest at trophozoite, reduction in merozoite number as well as effect on food vacuole functioning resulting in decreased haemozoin crystal. These could be generalised stress response and no direct correlation is established with the GDPD reduction. Why haemozoin crystal formation is hindered in the KO parasites, and how is it linked with disturbance in lipid metabolism and specifically PC synthesis?

Rescue of KO parasites with choline is intriguing but somewhat inexplicable. In the parasite, majority of PC is synthesised using choline derived from lysoPC, whereas under LPC depleted condition the parasite is shown to utilize PMT pathway for PC synthesis by compensating PE synthesis, rather than utilizing available choline in the serum. Even if some PC gets synthesized using choline taken from milieu, it should be able to recover parasite growth to only a certain level, however, in this study major recovery of parasite growth is shown. Mechanistic proof of choline rescue is missing. It is important to assess phospholipid profile to show normalized PC (and other phospholipids) synthesis under choline rescue condition. The authors should use labelled choline to show its incorporation in the PC synthesised under rescue condition. Can excess glycerophosphocholine (GPC) rescue the parasites under partial knock-down conditions?

Other comments:

– I did not get the supplementary figures on the website or in the BioRxiv site.

– Figure 4B: the parasitaemia is marked in log scale which is confusing to compare between different sets.

– Figure 5: Why LPCs are getting accumulated in the KO, which are targets of Lysophospholipase, whereas the direct target of GDPD is Glycero-phosphocholine. What is the status of glycero-phosphocholine in these parasites?

– Figure 5C: The labelled proportion of the majority of PC species is not significantly different between the two clones.

– The experimental setup for Figure 5C is confusing, the G1 parasite strain is expected to be completely devoid of PfGDPD, if the GDPD is only enzyme responsible for release of choline from GPC, then how can this parasite utilize lyoPC for labelling the newly synthesised PC.

– What is the significance of betaine lipid DGTS upregulation, DGTS is synthesised without the use of choline but using DAGs, how does this data correlate with other lipidomic profile, DAG levels as well as with the choline rescue.

*Reviewer #2 (Recommendations for the authors):*

1. Is GDPD localization to the PV and/or cytoplasm critical for parasite viability? Some discussion of this question seems warranted at a minimum. This key question could be strongly addressed if the authors extend the experiment in Figure 2G to test if episomal expression of GDPD variants either lacking the SP (and thus presumably just cytoplasmic) or lacking Met19 (which may prevent a cytoplasmic population and just retain the PV population) rescue parasites from loss of endogenous GDPD.

2. Do the GDPD mutations studied episomally in Figure 2G impact or reduce protein stability and expression? The microscopy data in the figure supplement suggest that all proteins are expressed but quantitative comparison is difficult. Can the authors show by western blot (normalized to loading control) that WT and enzyme mutants are all expressed at similar levels? This experiment seems critical since the 2012 Klemba study reported that a different active site mutant resulted in insoluble protein.

3. Is the mechanism of choline uptake into parasites known? Are New Permeability Pathway mechanisms involved (e.g., does furosemide antagonize uptake?)?

4. For the experiment in Figure 5A, does addition of exogenous choline to +RAP parasites lead to phospholipid levels that phenocopy -RAP conditions? This experiment may give insight into other GDPD functions beyond choline-dependent phospholipid synthesis.

5. The current labeling in Figure 5C is confusing and difficult to match to the relevant textual descriptions, as this same panel is referenced in lines 250 and 259 in different Results sections. The Figure 5C graph on the right (specific PC labeling) is also described in the text before the left graph, which adds confusion. It would be helpful to explicitly label the left graph as total lipids and possibly label the left/right graphs as distinct panels C and D that can be referred to separately and matched with the relevant text descriptions.

6. The 2012 Klemba paper previously showed that recombinant GDPD could hydrolyze glycerophosphocholine. Figure 6 in the present paper is a valuable extension of this experiment to endogenous GDPD from parasites, and these experiments are important to strongly support the proposed enzyme function in choline mobilization. Comparison of IP samples of GDPD {plus minus}RAP or to stable GDPD-deletion clone G1 provides some but incomplete support for specific function, as there could also be other functional losses in parasites that accompany GDPD deletion or co-purifying proteins that accompany GDPD pull-down that impact choline release in these experiments. Do anti-mCherry IPs under +RAP conditions of the parasites complemented with WT but not mutant enzymes (in Figure 2G) phenocopy GPC and choline levels in B4- parasites, as expected if differences between B4- and B4+ or G1 parasites are due to the specific activity of GDPD?

*Reviewer #3 (Recommendations for the authors):*

– The catalytic function of this protein has already been investigated in Denloye et al., 2012, this figure should be removed and instead the data from this previous article can be discussed.

– Not all defects, especially at the lipid level, can be explained with the current data and their interpretations: Why Do TAGs and DAGs increase in the mutant? Is there a link between accumulation of hemozoin crystals and neutral lipid accumulation, as recently reported (Asad et al. BMC Biology), have the author tested it? How do the authors explain that choline complementation rescues the parasite when this chline is believed to exclusively being generated from LPC?

– Some graphs don't have any error bars.

– Statistical significance is not always shown, and the tests performed are not discussed in the figure captions.

– Representation of the lipidomic data is often hard to follow, individual FA species of each lipid group is only shown in the supplementary figures, which also do not show the statistical significance of the data; were any stat tests done?

– No statistical significance is shown in Figure 5B and C.

– I would propose to the authors to replace the bubble charts that is quite difficult to read and interpret, with only the significant results of PE, PS, PC, LPC and DAG in the form of the bar graphs shown in Figure 5—figure supplement 1.

– In the PfGDPD-null mutant it would of interest to see if the expression of other transcripts has changed, and this can expand on how this parasite can survive solely on the exogenous choline despite having completely lost this proposed essential protein.

– Transcriptional analyses will also reveal why despite the LPC/PC balance being offset, sexual conversion is not induced.

– The lipidomic analyses is extremely detailed and it seems to be overly condensed to a single image and not fully deliberated in the discussion.

– What is the relation between the accumulation of DGTS and DAG? This could be addressed in the discussion of the paper.

---

## [Author Response]

Essential revisions:1) Provide additional panel in Figure 2 confirming localization of PfGDPH with a food vacuole marker2) Provide further evidence for how exogenous choline rescues GDPD KO parasite lipid levels, ideally via isotope tracing to unambiguously demonstrate utilization of exogenous choline rather than upregulation/utilization of an alternative source.3) Revise Figure 5 (especially 5C) to clarify the experimental set-up and better match to textual descriptions4) Discuss or confirm alternative possibilities for role of GDPH in gametocytogenesis

We thank the editor for this summary of the revisions considered essential for this manuscript. In the revised manuscript we have now dealt specifically with these 4 requests, as laid out in detail below. Note that we assume that for Essential Revision #1 above, the editor means the ‘parasitophorous vacuole’ rather than the ‘food vacuole’, which was not requested by the Reviewers.

Reviewer #1 (Recommendations for the authors):The gene knock out studies, as well as enzyme activity analysis of PfGDPD were carried out in an earlier study (Denloye et al., 2012), which highlighted its role in parasite survival and possible role in providing choline for phospholipid synthesis in the parasite.The phospholipid analysis shows a generalized response of disturbance in lipid homeostasis; phospholipid analysis in the first cycle (+RAP/-RAP) shows effect on levels of several phospholipids which are overall down, and a generalized increase in DAGs and TAGs. There is a clear mis-regulation of lipid metabolism in the GDPD-KO conditions, but it seems unclear how exactly it is linked with its proposed functions. The authors highlighted that the parasite can shift from CDP-choline pathway to the SDPM pathway to produce PC, which could be reason for lowered PE and PS production; however, the PS production is independent of PC or PE synthesis, which may suggest generalized effect on phospholipid profiles. Have the authors assessed upregulation of PMT pathway under KO conditions? What is the explanation for DAG and TAG increase under KO conditions?

We completely agree with the implied view of the Reviewer that our first lipidomics analysis, that of RAP+ and RAP- parasites at schizont stage in cycle 0 (original Figure 5A), whilst revealing a reduction in key structural phospholipids, does not reliably inform on the function of GDPD. While we chose this time point as a starting point for the lipidomics analysis for the reasons detailed (see lines 203-211 of the original submission), we initially underestimated the confounding effects of residual GDPD activity still present in the RAP+ parasites at the end of cycle 0 (original Figure 2D). Even those residual levels of GDPD can break down GPC to choline (Figure 6), likely explaining why our first lipidomics experiment revealed only a small and non-significant decrease in lysoPC-derived choline incorporation into PC (original Figure 5B). This, we suspect, combined with the compensatory SDPM pathway, resulted in unchanged PC levels in the cycle 0 schizonts. The ambiguity inherent in this experiment therefore motivated us to repeat the lipidomic analysis with the completely GDPD-null clonal line G1, following prolonged maintenance in culture (original Figure 5C), ultimately leading to a much clearer picture of GDPD function.

Although we did not directly assess upregulation of PMT activity in the GDPD-null G1 parasites, we believe that the reduction in PE and PS levels detected reflects activation of the SDPM pathway. This is because we detected a significant reduction in only PS, PE and π levels and not an overall decrease in all PLs (PC and PG are essentially unaffected). The reviewer is correct in stating that PS synthesis is independent of PC or PE synthesis under normal conditions, but upon activation of the SDPM pathway both serine and ethanolamine are expected to be redirected towards PC synthesis. Reflecting this, in the absence of lysoPC, serine contributes to 38.2% of PC biosynthesis in the parasite as shown by Wein *et al.* (2018).

Regarding levels of DAG and TAG in the GDPD-null parasites; in fact we observed increases in DAG but not in TAG levels (only one species of the detected TAG species was significantly upregulated; see Figure 5 —figure supplement 1). We suspect that the disturbance in precursor availability (choline, ethanolamine and serine) reduces DAG usage as the backbone for PL synthesis, resulting in its accumulation. We have now modified the revised manuscript (line 232 onwards) to specifically mention this possibility, as follows: “This disturbance in precursor availability likely reduces usage of DAG, the primary backbone for glycerolipid and neutral lipid production, resulting in its accumulation. Collectively, these results indicated the onset of disruption in PL biosynthesis and were consistent with a major role for PfGDPD in this process.”

In the cell biology study of the KO line, parasites show developmental arrest at trophozoite, reduction in merozoite number as well as effect on food vacuole functioning resulting in decreased haemozoin crystal. These could be generalised stress response and no direct correlation is established with the GDPD reduction. Why haemozoin crystal formation is hindered in the KO parasites, and how is it linked with disturbance in lipid metabolism and specifically PC synthesis?

We agree with the reviewer that the phenotypes described could simply result from a general stress response following GDPD ablation and the resulting decrease in PC levels. In the revised manuscript we have now modified line 326 to reflect this view, as follows: “Ablation of PfGDPD expression produced a phenotypic defect during trophozoite development likely resulting from a direct or general stress response to disrupted PC synthesis, similar to that observed upon inhibition of other enzymes of the CDP-choline pathway (Contet et al., 2015; Gonzalez-Bulnes et al., 2011; Serran-Aguilera et al., 2016).”

We note that Reviewer #3 also raised a similar question regarding haemozoin crystal formation. The reduced haemozoin formation could be a result of the stress response or due to the decrease in TAG levels we observe (original Figure 5C). Neutral lipids have been suggested to play a role in the parasite haem detoxification pathway and promote haemozoin formation (Hoang et al. 2010). Indeed it was recently shown that knockdown of a *P. falciparum* lysophospholipase results in reduced TAG levels, reduced haemozoin formation and a block in trophozoite development (Asad et al. 2021). It is therefore entirely plausible that the associated developmental defect is due to disruption in haem detoxification rather than directly by reduced PC levels. Since the TAG-haemozoin link is not well established in the current scientific literature (Matz JM et al., 2022), we previously refrained from drawing any further conclusions in our original submission. However, in view of this Reviewer’s comments (as well as those of Reviewer #3 – see below), we now briefly discuss this from line 383 in the revised manuscript – “The reduced haemozoin formation in GDPD-null parasites could also be a result of this decrease in TAG levels. Neutral lipids have been suggested to play a role in the parasite haem detoxification pathway and promote haemozoin formation (Hoang et al., 2010). Indeed it was recently shown that knockdown of a *P. falciparum* lysophospholipase results in reduced TAG levels, reduced haemozoin formation and a block in trophozoite development (Asad et al., 2021).”

Rescue of KO parasites with choline is intriguing but somewhat inexplicable. In the parasite, majority of PC is synthesised using choline derived from lysoPC, whereas under LPC depleted condition the parasite is shown to utilize PMT pathway for PC synthesis by compensating PE synthesis, rather than utilizing available choline in the serum. Even if some PC gets synthesized using choline taken from milieu, it should be able to recover parasite growth to only a certain level, however, in this study major recovery of parasite growth is shown. Mechanistic proof of choline rescue is missing. It is important to assess phospholipid profile to show normalized PC (and other phospholipids) synthesis under choline rescue condition. The authors should use labelled choline to show its incorporation in the PC synthesised under rescue condition. Can excess glycerophosphocholine (GPC) rescue the parasites under partial knock-down conditions?

It has been shown previously that exogenous choline rescues both wild-type and PMT-null parasites (i.e. lacking a functional SDPM pathway) that otherwise fail to survive (Witola et al., 2008, Wein et al., 2018). The reviewer is correct in observing that choline “should be able to recover parasite growth to only a certain level”. Consistent with that, while we see a significant rescue of parasite growth rate, choline does not completely restore wild-type growth rates (Figure 4B and 4E). This matches previous observations by others (Witola et al., 2008, Wein et al., 2018).

The mechanism underlying the rescue of GDPD-null parasites by choline is important and this question was also raised by Reviewers #2 and #3. A plausible explanation can be gleaned from prior work done in this area. LysoPC is generally the main source of choline for the parasite because at the normal choline concentration in human plasma (~10 uM), choline influx through the RBCM is very low so there is a bottleneck in transport of free choline across the red blood cell membrane (RBCM) (Biagini et al., 2004). However, at supraphysiological concentrations of choline as used in our study (500 uM), choline influx into the infected RBC is increased (Kirk et al., 1991). Incorporation of exogenous choline into PC has been previously demonstrated in lysoPC-starved parasites in Brancucci et al. (2017) using high levels (1mM) of d_9_-labelled exogenous choline. Thus, in the absence of lysoPC, the parasite can use exogenous choline to make PC so long as it is provided at relatively high concentrations.

The main question we address in our study is whether disrupting GDPD function mimics a lysoPC-deprived state, thereby implicating GDPD in lysoPC breakdown. We demonstrate this by metabolic labelling using a choline-labelled lysoPC species and show a 25-50% reduction in sourcing of choline from lysoPC in the absence of GDPD. As to the mechanistic proof of choline rescue of GDPD-null parasites, we maintain that we can rely on the previous studies (detailed above) that have reported both choline rescue and choline incorporation into PC in lysoPC-deprived parasites, and that repetition of those data using labelled choline is unnecessary. Instead, in our revised manuscript we have now discussed this issue further (lines 344 onwards), as follows: “Previous work has shown that supplying an excess of choline can rescue the growth of parasites in lysoPC-deprived conditions (Wein et al., 2018; Witola et al., 2008). High concentrations of exogenous choline partially surpass the bottleneck in choline transport across the red blood cell membrane (Biagini et al., 2004) and increase choline influx through infected RBCs (Kirk et al., 1991). LysoPC-deprived parasites can readily take up this choline and use it for PC synthesis as shown by previous metabolic labelling studies (Brancucci et al., 2017). The impact of exogenous choline was pronounced in our PfGDPD-null mutants as they completely failed to survive in the absence of choline but achieved near-wild type growth rates in high choline concentrations. This complete dependence on exogenous choline despite the abundant presence of lysoPC (amounting to ~17% of total lipid content (Garcia-Gonzalo and Izpisua Belmonte 2008)) in the Albumax II-supplemented culture media used in our studies or supplementation with GPC, provides further confirmation that ablating PfGDPD function interferes with choline acquisition from lysoPC and mimics a lysoPC-deprived state.” We trust that these modifications to our manuscript are acceptable to the Reviewer.

Supplying exogenous GPC does not rescue growth of the GDPD-null parasites. We now include these data in the revised submission (Figure 4C) and discuss them in lines 188 and 354.

Other comments:– I did not get the supplementary figures on the website or in the BioRxiv site.

We trust that this problem is now solved with the submission of our revised manuscript.

– Figure 4B: the parasitaemia is marked in log scale which is confusing to compare between different sets.

We appreciate the Reviewer’s point here but given the wide range of the parasitaemia levels in the -RAP and +RAP cultures, we feel it appropriate to present the data using a log scale to best visualise the differences between the choline-supplemented and control conditions. We hope this is acceptable.

– Figure 5: Why LPCs are getting accumulated in the KO, which are targets of Lysophospholipase, whereas the direct target of GDPD is Glycero-phosphocholine. What is the status of glycero-phosphocholine in these parasites?

The Reviewer raises an interesting point. In our experience, lysoPC measurements vary somewhat in our lipidomics data. This may be due to lysoPC in the Albumax II- supplemented media that may accumulate within the red blood cells (and that we could not get rid of entirely by washing the parasite pellet). As the Reviewer will see from the presented data, the differences in lysoPC levels referred to are not statistically significant due to wide inter-replicate variability. Therefore, we do not attribute much meaning to these “changes” in lysoPC levels.

Regarding the status of GPC in the parasites, this is a polar metabolite and therefore unfortunately was not captured by the lipid extraction and LC-MS methods that we employed in this work.

– Figure 5C: The labelled proportion of the majority of PC species is not significantly different between the two clones.

All species above and including PC(16:0_16:1) show significant reduction in labelled proportions in G1 parasites compared to the B4 controls (a 25-50% decrease with p value <0.05 from three replicates). We had previously shown absolute differences in the dot plot (G1-B4). We have now revised the dot plots shown in Figure 5B and C to present the same data as percentage change in labelling ((G1-B4)/B4 *100) to enable easier interpretation. The changes with p value <0.05 have also been now explicitly highlighted in the revised Figure 5C. Line 258 has also been altered to “However, a consistent and significant decrease (25-50%) in labelling of 10 out of 13 PC species was observed in the choline-starved PfGDPD-null G1 parasites compared to the B4 controls.” We hope these revisions are satisfactory.

– The experimental setup for Figure 5C is confusing, the G1 parasite strain is expected to be completely devoid of PfGDPD, if the GDPD is only enzyme responsible for release of choline from GPC, then how can this parasite utilize lyoPC for labelling the newly synthesised PC.

The reviewer raises an important point here. We also expected no labelling in the G1 parasites, but we suspect that alternate pathways such as direct acylation of lysoPC into PC via the Lands’ cycle could contribute to the labelled fraction in the PC pool. The enzymes involved in these pathways remain to be elucidated. We suggest this in lines 360 onwards of the revised manuscript, as follows: “This suggests that alternate pathways such as (i) direct lysoPC acylation into PC via the Lands’ cycle, (ii) two acylation steps to convert GPC to PC, or (iii) the SDPM pathway may contribute to PC synthesis under choline-starved conditions. However, our results strongly suggest that PfGDPD-mediated choline release from lysoPC remains the primary, indispensable pathway to meet the choline requirements of the intraerythrocytic parasite.” We hope that this text clarifies our interpretation of our observations.

– What is the significance of betaine lipid DGTS upregulation, DGTS is synthesised without the use of choline but using DAGs, how does this data correlate with other lipidomic profile, DAG levels as well as with the choline rescue.

We discuss at length the potential significance of DGTS in the Discussion (lines 389 onwards). Accumulation of DGTS as a non-choline/phosphorous substitute for PC in the choline-starved GDPD-null clonal G1 line is similar to that observed previously in fungi and marine bacteria under similar conditions. As mentioned earlier, due to the presence of residual GDPD levels and possibly activation of the SDPM pathway, RAP-treated cycle 0 GDPD:loxPint:HA schizonts are not severely choline-starved and their lipidomic profile (new Figure 5A) is not relevant here. However, extensive DAG accumulation is seen in both lipidomic profiles due to dysfunctional phospholipid metabolism and likely far exceeds any of its use in DGTS synthesis.

Reviewer #2 (Recommendations for the authors):1. Is GDPD localization to the PV and/or cytoplasm critical for parasite viability? Some discussion of this question seems warranted at a minimum. This key question could be strongly addressed if the authors extend the experiment in Figure 2G to test if episomal expression of GDPD variants either lacking the SP (and thus presumably just cytoplasmic) or lacking Met19 (which may prevent a cytoplasmic population and just retain the PV population) rescue parasites from loss of endogenous GDPD.

We appreciate the Reviewer’s experimental suggestions here, but respectfully suggest that they address questions that are beyond the remit of the present study. While the suggested experiments appear initially straightforward, they could well turn out to represent a substantial additional workload. The suggestions are based on the reasonable but as yet untested assumption that the internal Met19 is indeed a cryptic translation initiation site. Simple substitution of Met19 might interfere with the enzymatic function of the longer translation product, and different amino acid substitutions may have to be explored. The relative expression levels of the two isoforms (if they exist) may be important. However, we do agree that some discussion of this issue is warranted, and the revised manuscript now includes this on lines 404-411.

2. Do the GDPD mutations studied episomally in Figure 2G impact or reduce protein stability and expression? The microscopy data in the figure supplement suggest that all proteins are expressed but quantitative comparison is difficult. Can the authors show by western blot (normalized to loading control) that WT and enzyme mutants are all expressed at similar levels? This experiment seems critical since the 2012 Klemba study reported that a different active site mutant resulted in insoluble protein.

We accept this point. We have now quantified the mean GDPD-mCherry fluorescence intensity from a number of imaged parasites as a measure of GDPD expression. The results do not detect any statistically significant differences between the different lines. These data have now been added to the revised manuscript as Figure 2—figure supplement 3G. Line 152 of the manuscript has also been amended to read: “Mutagenesis of these key residues did not alter the expression or subcellular localization of the transgenic PfGDPD variants (Figure 2—figure supplement 3F and G) but completely abolished rescue of parasite growth upon disruption of the chromosomal gene (Figure 2G).”

3. Is the mechanism of choline uptake into parasites known? Are New Permeability Pathway mechanisms involved (e.g., does furosemide antagonize uptake?)?

Current understanding of the mechanism of choline uptake is explained above in our response to Reviewer #1. Choline indeed crosses the host red blood cell membrane via the New Permeability Pathway and into the parasite through a parasite choline carrier (Biagini et al., 2004). The revised manuscript now mentions this (line 72 onwards). Choline transport into the host RBC can be blocked by furosemide. However, furosemide does not affect choline uptake by the parasite via the choline carrier.

4. For the experiment in Figure 5A, does addition of exogenous choline to +RAP parasites lead to phospholipid levels that phenocopy -RAP conditions? This experiment may give insight into other GDPD functions beyond choline-dependent phospholipid synthesis.

This is an excellent suggestion and could be considered for future work. We did plan to include a sample corresponding to the G1 clone in the presence of choline for the lipidomics analysis shown in Figure 5C, but due to certain practical restraints we had to limit our analysis to the 2 sample groups shown.

5. The current labeling in Figure 5C is confusing and difficult to match to the relevant textual descriptions, as this same panel is referenced in lines 250 and 259 in different Results sections. The Figure 5C graph on the right (specific PC labeling) is also described in the text before the left graph, which adds confusion. It would be helpful to explicitly label the left graph as total lipids and possibly label the left/right graphs as distinct panels C and D that can be referred to separately and matched with the relevant text descriptions.

We appreciate these suggestions, and in the revised manuscript the figure has now been revised accordingly to make it easier to relate to the relevant text.

6. The 2012 Klemba paper previously showed that recombinant GDPD could hydrolyze glycerophosphocholine. Figure 6 in the present paper is a valuable extension of this experiment to endogenous GDPD from parasites, and these experiments are important to strongly support the proposed enzyme function in choline mobilization. Comparison of IP samples of GDPD {plus minus}RAP or to stable GDPD-deletion clone G1 provides some but incomplete support for specific function, as there could also be other functional losses in parasites that accompany GDPD deletion or co-purifying proteins that accompany GDPD pull-down that impact choline release in these experiments. Do anti-mCherry IPs under +RAP conditions of the parasites complemented with WT but not mutant enzymes (in Figure 2G) phenocopy GPC and choline levels in B4- parasites, as expected if differences between B4- and B4+ or G1 parasites are due to the specific activity of GDPD?

We agree with the reviewer that the earlier demonstration by Denloye et al., 2012 that recombinant GDPD hydrolyses GPC to produce G3P was important, and we were very happy to be able to complement those data in our study by demonstrating that native, parasite-derived enzyme can generate free choline from GPC. Equally importantly, our approach also enabled us to show that even the residual amounts of GDPD in RAP-treated cycle 0 schizonts display measurable catalytic activity, nicely explaining the mild phenotype of the parasites at that point in their development.

We appreciate the Reviewer’s reservations regarding the fact that our pull-down experiments do not completely rule out a role for co-purifying proteins in the GPC hydrolysis we observed. Co-purification of other proteins that contribute to GPC hydrolytic activity is a formal possibility but arguing against this, we did not detect any additional Coomassie-stained bands other than PfGDPD in the SDS PAGE analysis of our pull-down fractions (Figure 6—figure supplement 2). Furthermore, our complementation experiments with WT and mutant mCherry-tagged GDPD show unambiguously that parasite viability is not supported by mutant enzymatically inactive PfGDPD, which might still form interactions with partner proteins. The reviewer’s suggestion is a good one to rule out a possible enzymatic role (or co-factor role) for partner proteins, but we consider this unlikely in the light of knowledge of other GDPD enzymes, which generally function without protein co-factors. From a practical perspective, there is substantial expense associated with the mass-spectrometric analysis required to show hydrolysis to free choline in our assays. For all these reasons, we have not performed the proposed experiments using anti-mCherry to pull-down mCherry-tagged GDPD from parasites complemented with such constructs. We hope this is acceptable.

Reviewer #3 (Recommendations for the authors):– The catalytic function of this protein has already been investigated in Denloye et al., 2012, this figure should be removed and instead the data from this previous article can be discussed.

It is unclear which figure the reviewer refers to here. If it is Figure 6, we agree with the reviewer that the catalytic function of recombinant GDPD has been partially characterised previously by Denloye et al., 2012. In our revised manuscript we have now added a sentence (line 292) specifically mentioning this, as follows:

“Recombinantly expressed PfGDPD has previously been shown to possess magnesium-dependent hydrolytic activity that releases G3P from GPC (Denloye et al., 2012)”. However, in agreement with the view of Reviewer #2 (see above) we believe it is important to retain our experimental data in the current manuscript for three reasons. First, we confirm the hydrolytic activity of GDPD using a protocol that is completely different from Denloye et al., 2012, in that our in vitro assays used PfGDPD isolated directly from parasites as opposed to the recombinant protein used previously. Second, we used LC-MS to directly quantify GPC and choline in the reactions, whereas Denloye et al., 2012 measured G3P production using a two-step assay as a measure of GDPD activity. Finally – and importantly – our data confirm that even the residual amounts of GDPD in our RAP-treated cycle 0 parasites display measurable, choline releasing catalytic activity. This was crucial to explain the phenotype and lipidomics profile of cycle 0 schizonts. We trust that the reviewer accepts our reasoning on this issue.

– Not all defects, especially at the lipid level, can be explained with the current data and their interpretations: Why Do TAGs and DAGs increase in the mutant? Is there a link between accumulation of hemozoin crystals and neutral lipid accumulation, as recently reported (Asad et al. BMC Biology), have the author tested it? How do the authors explain that choline complementation rescues the parasite when this chline is believed to exclusively being generated from LPC?

We agree with the reviewer that there could indeed be a link between hemozoin generation and neutral lipid accumulation; please see our response above to Reviewer #1. Much further experimentation will be required to confirm this link and we suggest that this is beyond the remit of this current work.

The second point raised by this Reviewer is also mentioned above in our response to Reviewer #1. Briefly, lysoPC is the preferred source of choline for the parasite. However, high concentrations of exogenous choline partially surpass the bottleneck in choline transport across the red blood cell membrane (Biagini et al., 2004) and increase choline influx through the infected RBC (Kirk et al., 1991). LysoPC-deprived parasites readily take up this choline and use it for PC synthesis, as previously shown by metabolic labelling studies (Brancucci *et al.*, 2017).

– Some graphs don't have any error bars.– Statistical significance is not always shown, and the tests performed are not discussed in the figure captions.– Representation of the lipidomic data is often hard to follow, individual FA species of each lipid group is only shown in the supplementary figures, which also do not show the statistical significance of the data; were any stat tests done?– No statistical significance is shown in Figure 5B and C.– I would propose to the authors to replace the bubble charts that is quite difficult to read and interpret, with only the significant results of PE, PS, PC, LPC and DAG in the form of the bar graphs shown in Figure 5—figure supplement 1.

Statistical significance was calculated for all the data. As stated in the methods, in all cases – “Student’s t-test were used to compare group means and where necessary Bonferroni adjustment for multiple comparisons was applied to the p-value of statistical significance.”

Bar graphs in Figure 5 represent the average labelled proportion in each parasite line. Their variability has been shown as error bars in the dot plots on the side that denote the differences between the proportions. Statistical significance is now shown in the revised Figure 5C for labelling data.

In the main Figure 5C bubble plot, a scale has been provided depicting the p-value associated with each point. Bubbles bigger than the size denoted are statistically significant. In the supplementary figure (Figure 5—figure supplement 1), ONLY the statistically significant species are displayed (as described in the figure legend). We would like to retain the bubble charts as they provide the reader with an eagle-eye’s view of two lipidomic experiments within the space of a single figure and inform on the main changes we discuss in the text. The reader can always then refer to the supplementary bar graphs. We trust this is acceptable.

– In the PfGDPD-null mutant it would of interest to see if the expression of other transcripts has changed, and this can expand on how this parasite can survive solely on the exogenous choline despite having completely lost this proposed essential protein.– Transcriptional analyses will also reveal why despite the LPC/PC balance being offset, sexual conversion is not induced.

We appreciate these suggestions, and indeed we did initially consider bulk RNAseq analysis of the mutants. However, due to the severe developmental defect we observed in the mutants (in both the 3D7 and NF54 lines), we felt there would be wide transcriptional variations across the board that would likely confound any meaningful changes arising from PfGDPD ablation. Furthermore, since provision of high concentrations of exogenous choline likely merely replaces choline that the parasite would have derived from breakdown of lysoPC, we doubt that there is any alternate pathway to be elucidated through transcriptomics. As described above in our response to Reviewer #2, we suspect that the lack of sexual conversion in the GDPD-null NF54 parasites is simply because the parasites die (due to a choline deficiency) before they can progress to gametocytogenesis.

– The lipidomic analyses is extremely detailed and it seems to be overly condensed to a single image and not fully deliberated in the discussion.

We are glad the reviewer appreciates our lipidomics efforts. Since we also have a large amount of phenotypic and biochemical data (the manuscript already contains 7 figures), we had no choice but to condense the lipidomic work into a themed figure. We have tried to be both informative and concise in our Discussion.

– What is the relation between the accumulation of DGTS and DAG? This could be addressed in the discussion of the paper.

We discuss significance of DGTS in the Discussion (Lines 383-396). While DGTS is made from DAG, a good degree of DAG accumulation is seen in both lipidomic profiles due to dysfunctional phospholipid metabolism and this would far exceed any of its use in DGTS synthesis.